# TS-Reasoner: Domain-Oriented Time Series Inference Agents for Reasoning and Automated Analysis

**Wen Ye**[*]                                                                 *yewen@usc.edu*
*Department of Computer Science*
*University of Southern California*

**Wei Yang**[*]                                                              *wyang930@usc.edu*
*Department of Computer Science*
*University of Southern California*

**Defu Cao**                                                                  *defucao@usc.edu*
*Department of Computer Science*
*University of Southern California*

**Yizhou Zhang**                                                    *yizhou.zhangyiz@usc.edu*
*Department of Computer Science*
*University of Southern California*

**Lumingyuan Tang**                                                        *lumingyu@usc.edu*
*Department of Computer Science*
*University of Southern California*

**Jie Cai**                                                                        *caijie@usc.edu*
*Department of Computer Science*
*University of Southern California*

**Yan Liu**                                                                    *yanliu.cs@usc.edu*
*Department of Computer Science*
*University of Southern California*

**Reviewed on OpenReview:** *https://openreview.net/forum?id=yhy7Vigjcf*

## Abstract

Time series analysis is crucial in real-world applications, yet traditional methods focus on isolated tasks only, and recent studies on time series reasoning remain limited to either single-step inference or are constrained to natural language answers. In this work, we introduce TS-Reasoner, a domain-specialized agent designed for multi-step time series inference. By integrating large language model (LLM) reasoning with domain-specific computational tools and error feedback loop, TS-Reasoner enables domain-informed, constraint-aware analytical workflows that combine symbolic reasoning with precise numerical analysis. We assess the system's capabilities along two axes: 1) fundamental time series understanding assessed by TimeSeriesExam and 2) complex, multi-step inference, evaluated by a newly proposed dataset designed to test both compositional reasoning and computational precision in time series analysis. Experiments show that our approach outperforms standalone general-purpose LLMs in both basic time series concept understanding as well as the multi-step time series inference task, highlighting the promise of domain-specialized agents for automating real-world time series reasoning and analysis.

---

[*]Equal contribution. Please see `https://github.com/wen-ye-xwz/TS-Reasoner` for code and dataset release.

# 1 Introduction

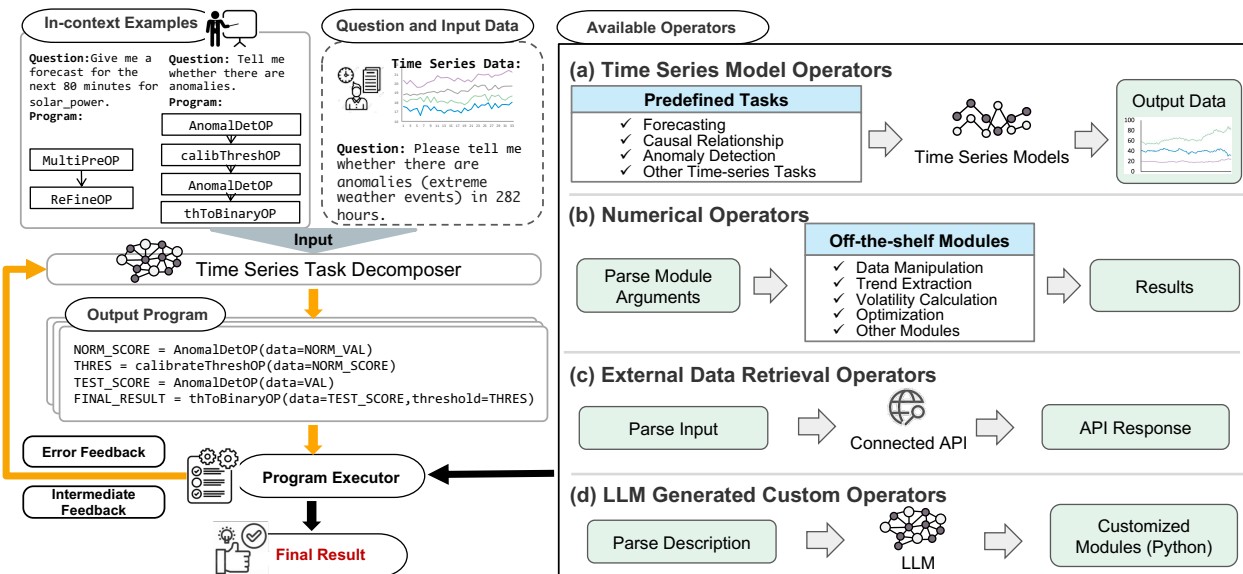

Figure 1: The architecture of TS-Reasoner. The LLM work as task decomposer, which learn from operator definitions and in-context examples to decompose task instances into sequence of operators. Then a program executor is responsible for solution plan execution and forms a feedback loop with task decomposer.

Time series analysis lies at the heart of many high-impact real-world applications, powering decisions in domains such as finance, healthcare, and energy systems (Cheng et al., 2021; Sharma et al., 2021; Zhang et al., 2021). Over the past decades, research in time series modeling has seen significant advancements, not only in canonical tasks such as forecasting, anomaly detection, and classification (Hamilton, 2020; Lim & Zohren, 2021; Ismail Fawaz et al., 2019; Zamanzadeh Darban et al., 2024), but also in the development of foundation models (Ansari et al., 2024; Liu et al., 2024b; Garza & Mergenthaler-Canseco, 2023; Goswami et al., 2024). These models enable zero-shot inference and achieve strong performance on benchmarked tasks, substantially expanding the scope of data-driven time series analysis.

Despite this progress, real-world time series analysis rarely conforms to a fixed input–output paradigm. Practical inference often requires procedural reasoning: retrieving auxiliary covariates, invoking appropriate forecasting or diagnostic tools, validating operational constraints, and iteratively refining intermediate results. For instance, electricity load prediction in power grid operations typically involves combining exogenous data, applying forecasting models, checking system-level constraints, and adjusting predictions accordingly (WANG et al., 2016). Such workflows are inherently multi-step and compositional, yet are not captured by existing models or benchmarks.

This gap motivates a new task setting that we term multi-step time series inference, in which solving a single query requires dynamically composing multiple analytical steps, numerical tools, and domain constraints. While time series foundation models excel at individual subtasks, they lack mechanisms for instruction following, workflow assembly, and constraint-aware reasoning. Large language models (LLMs), on the other hand, exhibit strong reasoning and instruction-following capabilities (Mondorf & Plank, 2024; Zeng et al., 2023), but struggle with temporal pattern recognition and numerical computation due to discretization artifacts (Spathis & Kawsar, 2024) and their optimization for token prediction rather than numerical estimation (Li et al., 2024; Cvejoski et al., 2022; Selvam, 2025). Addressing these limitations through full retraining or architectural redesign is costly and inefficient.

In this work, we advocate a domain-oriented specialization of LLMs for time series inference via program-aided execution. We introduce TS-Reasoner, a time series domain-specialized agent that decomposes high-level inference queries into structured workflows composed of well-defined time series operators, ensuring

numerical accuracy and reliable execution. Rather than embedding time series reasoning within the LLM itself, TS-Reasoner leverages in-context learning (Brown et al., 2020) to learn how to orchestrate domain-specific operators for multi-step inference. Standard execution feedback is used as an enabling mechanism to iteratively refine generated workflows when errors occur, improving robustness without retraining.

Robust evaluation of multi-step time series inference requires benchmarks that go beyond single-step prediction accuracy. We first assess TS-Reasoner's foundational time series understanding using TimeSeriesExam (Cai et al., 2024). We then introduce a new benchmark composed of curated tasks that reflect recurring challenges in scientific and engineering practice, including constraint-aware forecasting in smart grid operations (Han et al., 2021; WANG et al., 2016; Greif et al., 1999), anomaly detection with threshold calibration (Yan et al., 2021), and causal discovery guided by domain knowledge(Aoki & Ester, 2020) (instantiated through a Granger-style predictive notion of temporal influence(Seth, 2007)). These tasks serve as representative instances of multi-step time series inference, capturing the procedural and analytical complexity absent from existing benchmarks.

We summarize our contributions as follows: (1) We formalize multi-step time series inference as a task paradigm that captures procedural reasoning, constraint handling, and domain knowledge integration, which are largely missing from existing models and benchmarks. (2) We propose TS-Reasoner, a domain-oriented time series agent that performs multi-step inference by composing structured workflows over time series operators, ensuring numerical reliability through programmatic execution. (3) We construct a realistic benchmark grounded in scientific and engineering domains, and demonstrate that TS-Reasoneroutperforms general-purpose LLM agents while revealing their limitations in time series settings.

## 2 Related Work

**Traditional Time Series Tasks and Foundation Models**  Classical time series analysis encompasses key tasks such as forecasting (Ekambaram et al., 2023), imputation (Tashiro et al., 2021), classification (Zhao et al., 2017), and anomaly detection (Xu et al., 2021), each serving unique purposes across domains. Traditionally, these tasks were addressed by dedicated models optimized for specific objectives, resulting in a fragmented modeling landscape. Inspired by advances in LLMs, recent work has proposed general-purpose time series foundation models (Woo et al., 2024; Liu et al., 2024b). LLMTime (Gruver et al., 2024) encodes time series as strings, while TimeLLM (Jin et al., 2023) and TimeMoE (Shi et al., 2025) align time series with language representations. TEMPO (Cao et al., 2023) combines decomposition and prompt design to enable generalization to unseen data, and Chronos (Ansari et al., 2024) leverages scaling and quantization for efficient embedding. While these models handle multiple preset tasks within a unified architecture, they remain constrained by fixed task definitions and lack support for procedural, compositional inference involving dynamic task composition and constraint reasoning.

**LLM Reasoning & LLM-Based Agents**  Large Language Models (LLMs) exhibit strong reasoning capabilities when aided by in-context learning and structured prompting (Huang & Chang, 2022; Qiao et al., 2022; Ahn et al., 2024). Techniques such as Chain-of-Thought prompting and its extensions (Wei et al., 2022; Yao et al., 2023a;b; Wang et al., 2022; Qu et al., 2024) encourage explicit intermediate reasoning, while program-based reasoning frameworks (Khot et al., 2022; Creswell & Shanahan, 2022; Gao et al., 2023) integrate external modules for precise execution, as exemplified by VisProg (Gupta & Kembhavi, 2023) in the vision domain. To address domain-specific limitations, recent agent systems incorporate structured tool interfaces and feedback mechanisms (Sun et al., 2023; Cai et al., 2025). Examples include SWE-agent (Yang et al., 2024) for software engineering, HoneyComb (Zhang et al., 2024) for materials science, and CRISPR-GPT (Huang et al., 2024) for biological sequence editing. Within the time series domain, recent works explore LLM-based agents for incorporating external context or event information into forecasting (Wang et al., 2024b; Lee et al., 2025). These approaches primarily target specific predictive tasks and model architectures. In contrast, TS-Reasoner focuses on multi-step time series inference as a task paradigm, emphasizing a domain-oriented operator toolkit and structured execution interface that supports heterogeneous tasks and constraint-aware workflows. More broadly, our work aligns with the Agent-as-Data-Analyst paradigm (Tang et al., 2025), and can be viewed as a time-series–specific instantiation with dedicated operators and evaluation protocols.

## 3 TS-Reasoner

In this section, we formally introduce TS-Reasoner which addresses the limitations of standalone LLMs in handling numerical precision and computational complexity in time series analysis by decomposing complex time series inference tasks into structured operator execution pipelines. The core architecture of TS-Reasoner can be expressed as:

$$R^i = \text{TaskDecomposer}(x, C, \text{feedback}^{i-1}) \tag{1}$$

$$\text{Exec}(R^i) = \begin{cases} y, & \text{if successful and valid} \\ \text{feedback}^i, & \text{if failed or poor quality} \end{cases} \tag{2}$$

where $x$ denotes the input task described in natural language, $C$ represents the in-context examples containing operator definitions and demonstrated operator usages in response to sample questions, $R^i = [f_1, f_2, ..., f_n]$ is the reasoning trace consisting of specialized operators planned by the task decomposer at iteration $i$, $y$ represents TS-Reasoner's final output obtained from successful execution of $R$. Figure 1 provides an overview of TS-Reasoner. The framework follows a modular design philosophy, separating high-level reasoning from computational execution. By delegating computationally intensive or mathematically precise operations to specialized operators.

**Task Decomposer** The task decomposer is embodied by a LLM and is responsible for transforming complex natural language instructions into structured operator execution plans. Given the definitions of operators in the toolbox and a set of example task-solution pairs $(x_1, R_1), (x_2, R_2), ..., (x_n, R_n)$, TS-Reasoner learns to map new tasks to appropriate decomposition sequences imitating human breaking down complex problems into manageable subtasks. The in-context learning paradigm does not require expensive retraining and allows LLM to recognize structural patterns in time series analysis workflows.

**Specialized Operators** TS-Reasoner relies on a suite of modular operators that wrap existing statistical tools, machine learning and deep learning models, and numerical utilities. We adopt state-of-the-art or widely used components and expose them through a unified interface. Each operator is given an informative name (e.g., ForecastOP, AnomalDetOP, calibrateThreshOP) and follows a consistent calling structure, enabling seamless orchestration within the execution pipeline. This standardized abstraction allows the task decomposer to compose complex workflows from heterogeneous tools in a model-agnostic and reusable manner. The available operators can generally be categorized into four groups, each addressing distinct aspects of time series analysis. **1) Time Series Model Operators** These operators form the analytical core of TS-Reasoner, applying machine learning and deep learning models to time series data. They handle canonical time series analysis tasks such as forecasting, imputation, and anomaly detection using models like ARIMA (Box & Jenkins, 1968), Chronos (Ansari et al., 2024), Lag-Llama (Rasul et al., 2023), and MOMENT (Goswami et al., 2024). Depending on the task, the amount of input data, and time series model choice, this family of operators can work in zero-shot mode or finetuning mode. **2) Numerical Operators** These provide standard statistical and mathematical functions used throughout the solution pipeline as well as simple shape manipulation operators to ensure seamless passing of variables. For example, operators within this family are equipped with functionalities like computing averages, performing decomposition, measuring volatility, or evaluating distribution properties. **3) External Data Retrieval Operators** These operators fetch relevant contextual data from external sources. The implementation of these operators mainly involves wrapping API calls to available services that supply time series based on queries. **4) LLM-Generated Custom Operators** To address the impracticality of predefining all necessary time series tools, we introduce a LLM-generated custom operator. This is a single operator that accepts a natural language prompt and returns executable code tailored to the specific analytical requirement, enabling a flexible and extensible pipeline that adapts to diverse task demands. This operator is used during refinement when task-specific post-processing logic is required beyond predefined operators (e.g., constraint-driven adjustment in electricity forecasting).The current toolkit covers 57 operators, a complete list of operators within each category is outlined in section C.

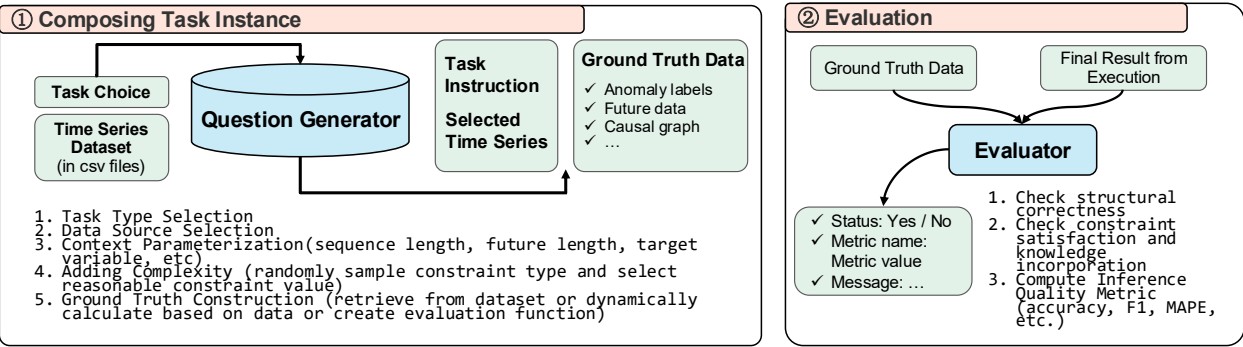

Figure 2: The proposed pipeline for multi-step time series inference task generation and evaluation.

**Feedback Mechanism for Self-Refinement**   Once the task decomposer generates a structured execution plan $R = [f_1, f_2, ..., f_n]$ that specifies the sequence and order of operators, the program executor interprets and executes each step accordingly. If an execution error is encountered, a feedback loop is initiated: the task decomposer is re-invoked to produce a revised end-to-end execution plan using the original query, in-context examples, the previously generated solution, and the error message from the current execution attempt (as shown in Equation 2). This error correction process is allowed to retry up to $k$ times, where $k$ is a predefined maximum number of retry attempts.

With the abundant suite of available time series foundation models, using time series model operators often necessitates model selection. In realistic settings, an intelligent agent may benefit from intermediate feedback either from automated evaluators or human/external signals to refine its model choices. Conditioned on such support from the evaluation benchmark, TS-Reasoner leverages feedback on prediction quality (e.g., MAPE) to experiment with different model choices and select the best-performing one. Feedback is obtained via within-instance, temporally valid self-evaluation, where the last 10% of each time series is used as a pseudo-holdout for intermediate feedback, while final evaluation is conducted exclusively on the unseen future horizon. The program executor maintains a buffer memory of previously explored solution paths to prevent redundant evaluations. If the task decomposer reverts to a previously attempted solution, the program execution proceeds directly to completion. This architecture ensures error handling, and adaptive refinement, enabling TS-Reasoner agent to dynamically optimize multi-step time series inference.

# 4   Multi-Step Time Series Inference Dataset

To study multi-step time series inference in realistic settings, we introduce a benchmark dataset centered on a task formulation that goes beyond single-step prediction. The dataset is designed to evaluate whether a system can decompose a natural-language instruction into multiple reasoning stages, perform time series analysis, incorporate domain knowledge, and enforce explicit constraints during inference. Unlike traditional time series benchmarks that focus on a single task such as forecasting accuracy, our benchmark comprises diverse task types, including constraint-aware forecasting, diagnostic analysis, and Granger-style causal reasoning. Each instance is framed as a natural-language instruction paired with context-rich time series inputs and well-defined evaluation criteria, enabling systematic assessment of both solution validity and inference quality.

We categorize the tasks into two broad classes: predictive tasks and diagnostic tasks. 1) Predictive tasks focus on time series forecasting under realistic operational constraints. These include requirements such as limiting the maximum allowable load in a grid, enforcing electricity load ramp rate restrictions, or bounding variability within an acceptable range (Greif et al., 1999; Oreshkin et al., 2020). Such constraints are common in engineering and energy systems, where violations may result in equipment damage, service disruption, or regulatory penalties (Nti et al., 2020). Our dataset simulates scenarios where future values must not only be predicted accurately but also satisfy operational constraints or leverage domain knowledge. To reflect practical challenges, we include forecasting scenarios with and without covariates data supplied in

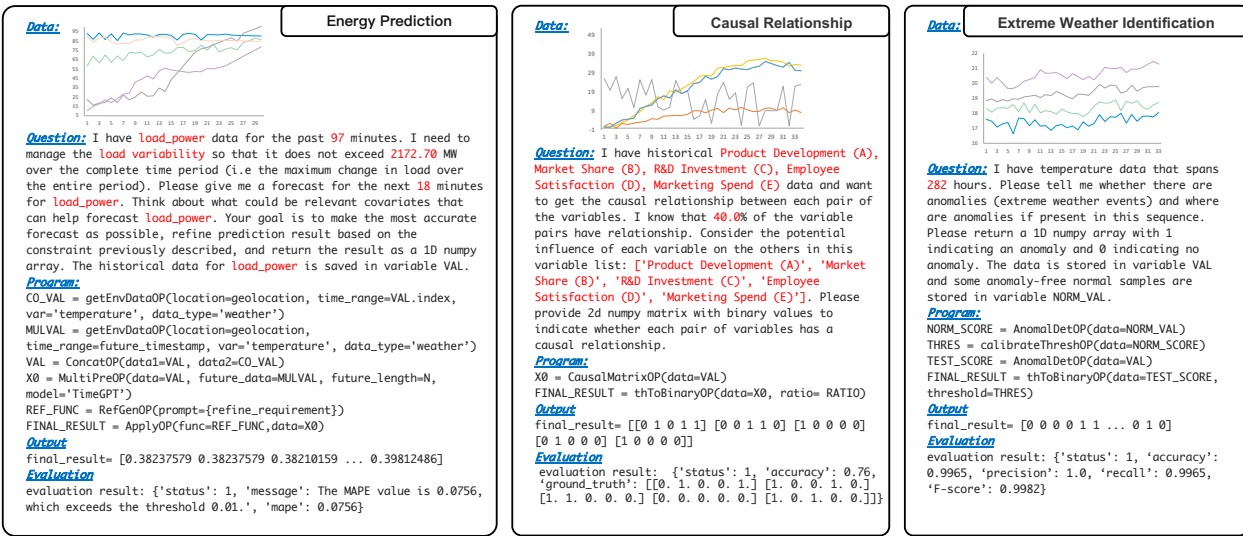

Figure 3: Examples of End-to-End Tasks on Time Series. These tasks require the model to perform multi-step reasoning on time series data. Each solution consists of a transparent sequence of steps, making the reasoning process significantly more interpretable than end-to-end multimodal models.

the question, and also scale to multi-grid settings where large volumes of data must be processed across multiple regions. 2) Diagnostic tasks emphasize pattern recognition, calibration, and structural inference. One subclass involves anomaly detection, where the model must identify abnormal patterns in a time series, often using reference samples or contextual priors such as known anomaly rates or anomaly-free samples to calibrate thresholds (Liu et al., 2016). These tasks simulate settings like extreme weather detection, where defining anomaly boundaries is not straightforward and depends on prior data distributions. Another subclass of diagnostic tasks centers on causal discovery. Here, the model is asked to infer Granger causal relationships among variables, given partial domain knowledge such as the expected proportion of causal links (Aoki & Ester, 2020). These tasks require the integration of weak supervision with statistical reasoning (Huang et al., 2020) and offer a rich testbed.

Figure 2 shows the pipeline used to generate proposed pipeline. Each task instance is generated via a modular pipeline grounded in real-world multi-step time series analysis. We first survey existing literature to identify three task definitions that demand multi-step inference: constraint-aware forecasting, extreme weather detection with known prior and causal discovery with domain knowledge. These are annotated with natural language templates reflecting their problem definition and objectives. To instantiate a task, we source time series data from datasets from corresponding domains: ERA5[1] (climate), PSML (Zheng et al., 2021) (electricity), and synthetic generators (causal graphs). Each specific task instance is parameterized by selecting time ranges, target variable, and context-specific constraints/knowledge. The result is a dynamic task specification with varying evaluation criteria across instances, distinguishing from traditional fixed-protocol benchmarks.

Evaluation is conducted through a unified but flexible protocol. Each task instance is paired with a dedicated evaluator configured with access to ground truth data, contextual information, and task-specific constraints or domain knowledge. The evaluator assesses the final model output along three dimensions: structural correctness (e.g., output shape and format), constraint satisfaction (e.g., anomaly rates, causal priors, load limits), and inference quality using appropriate metrics such as accuracy, F1 score, or MAPE. This framework supports rigorous and interpretable assessment across diverse task types. Such dedicated evaluators also support generating intermediate feedback quality signals like MAPE by comparing predictions to ground truth, simulating expert or human feedback in realistic decision-making workflows.

---

[1]https://climatelearn.readthedocs.io/en/latest/user-guide/tasks_and_datasets.html#era5-dataset

### 4.1 Task Instance Templates

In this section, we provide an outline of templates used for each type of tasks. The exact template for each sub question type may vary from each other to best reflect the available information: with and without covariates data supplied, whether multiple time series is supplied to test larger-scale multi-step inference cases, what known priors are provided (anomaly rate or reference free sample or known ratio of causal links), and which constrains are selected (global electricity load variability limit, maximum alloable load, e.t.c). The current scale of the proposed benchmark consists of around 500 specific task instances. Figure 3 demonstrates sample questions generated from the task instance generation pipeline.

**Predictive Tasks** In predictive tasks, we primarily focus on load-related issues in the energy sector. Specifically, each test sample provides the model with a natural language question, relevant time series historical data, and operational constraint requirements. The questions are generated by the following templates:

> **Question Template (Electricity Load Prediction)** *I have historical {influence variables} data and the corresponding target variable data for the past {historical length} minutes. [1. I need to ensure that the maximum allowable system load does not exceed {load value} MW. 2. I require that the system load is maintained above a minimum of {load value} MW. 3. I must monitor the load ramp rate to ensure it does not exceed {constraint value} MW for each time step. 4. I need to manage the load variability so that it does not exceed {constraint value} MW over the given period.] Think about how {influence variables} influence {target variable}. Please give me a forecast for the next {future length} minutes for target variable. Your goal is to make the most accurate forecast as possible, refine prediction result based on the constraint previously described. {output requirement: E.g. please return a 1D numpy array}.{data variable name specification: E.g. The historical data is stored in variable VAL.} .*

**Diagnostic Tasks** In diagnostic tasks, we primarily used extreme weather detection scenarios in climate science and synthetic Granger causal discovery. Specifically, each task instance includes natural language description of the question, time series, and domain knowledge description. The extreme weather detection questions are generated by the following template:

> **Question Template (Extreme Weather Detection)** *I have 2m temperature data that spans {sequence length} hours. Please tell me whether there are anomalies (extreme weather events) and where are anomalies if present in this sequence. [1. I also have some anomaly-free 2m temperature data from the same region. 2. I know that {anomaly ratio} percent of the times have anomalies. ] {output requirement}.{data variable name specification} .*

For Granger causal analysis, we synthesize a set of multivariate time series grounded in domain knowledge from climate science, finance, and economics. Specifically, we generate time series data according to prede-fined temporal influence structures, where directed relationships are defined under a Granger-style predictive notion of causality (Seth, 2007), consistent with common observational time series settings. Each test sample consists of a multivariate time series dataset accompanied by a natural language instruction asking the model to infer directed temporal dependencies among variables, along with expert-provided statistics on the proportion of true relationships. The reasoning model is required to infer dependencies based on the observed data and instructions, without access to the underlying generative process. The evaluation framework then measures performance by comparing the inferred dependency structure against the synthetic ground truth, enabling controlled and automatic assessment of causal reasoning within multi-step inference workflows. The questions are generated by the following template:

> **Question Template (Granger Causal Discovery)** *I have historical {variable names} data and want to get the causal relationship between each pair of the variables. I know that {ratio}% of the variable pairs have relationship. Consider the potential influence of each variable on the others in this variable list: {variable names}. {output requirement}.{data variable name specification}*

## 4.2 Dataset Statistics

Table 1 summarizes the dataset compiled for the multi-step time series inference task. The energy data with covariates contained electricity load from 6 major electricity grids in the United States (MISO, ERCOT, CAISO, NYISO, PJM, SPP) across 66 zones for 3 complete years (2018 -2020) with a minute level frequency [2]. The energy data without covariates contain both geolocation and load data for sub-regions within electricity grids at an hourly frequency. The data is retrieved from official websites of ERCOT, MISO, NYISO[3][4][5]. The climate data consisted of the ERA5 reanalysis dataset which is the most popular climate dataset. The extreme weather detection dataset is obtained from running scripts in climatelearn scripts[6].

## 5 Experiments

In this section, we conduct a series of comparative experiments to evaluate the performance of various LLM agents on both fundamental and complex time series understanding tasks. The evaluation includes multiple-choice questions assessing basic time series concepts, as well as more advanced inference tasks supported by the proposed dataset. Our baseline models include proprietary LLMs: GPT-4o (Hurst et al., 2024), DeepSeek model (Liu et al., 2024a), reasoning model GPT-o1 (Jaech et al., 2024).Additionally, we assess the ReAct agent based on GPT-4o (Yao et al., 2022), which synergizes reasoning and acting by generating traces for each in an interleaving manner. TS-Reasoner uses GPT-4o (Hurst et al., 2024) as the task decomposer. For all experiments, we perform a single run using the same hyperparameters for the most deterministic LLM output decoding: temperature=0.0 and top_p=1.0 for the most deterministic output. All experiments are ran using a single NVIDIA A40 GPU with 48G memory.

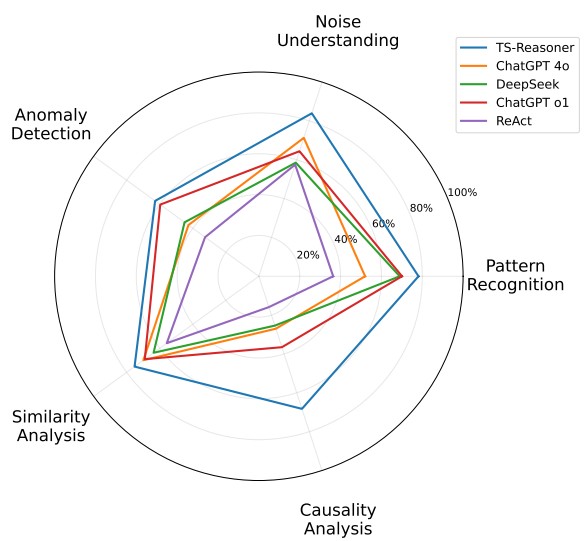

Figure 4: Strict Accuracy of TS-Reasoner and general purpose LLMs on the TimeSeriesExam.

### 5.1 Basic Time Series Understanding

To evaluate the fundamental capabilities of these models, we utilize the publicly available TimeSeriesExam benchmark (Cai et al., 2024) consisted of multiple choice questions. This benchmark assesses understanding across five key time series concepts: Anomaly Detection, Pattern Recognition, Noise Understanding, Causality Analysis, and Similarity Analysis. We use strict accuracy as evaluation metrics where only the final answer within the response of LLM is checked

---

[2]https://github.com/tamu-engineering-research/Open-source-power-dataset

[3]https://www.nyiso.com/load-data

[4]https://www.ercot.com/gridinfo/load/load_hist

[5]https://www.misoenergy.org/markets-and-operations/real-time--market-data/market-reports

[6]https://climatelearn.readthedocs.io/en/latest/user-guide/tasks_and_datasets.html#era5-dataset

| Dataset | Number of CSVs | Avg Total Timestamps | Number of Variables |
|---|---|---|---|
| Climate Data | 624 | 526 | 2048 |
| Energy Data | 22 | 8760 | 1-3 |
| Energy Data w/ Covariates | 66 | 872601 | 11 |
| Causal Data | 8 | 529 | 3–6 |

Table 1: Dataset Statistics of the constructed dataset. The exact number of time series are not calculated because it depends on randomly sampled sequence length when generating task instances.

against the ground truth answer instead of the complete response [7]. As illustrated in Figure 4, TS-Reasoner consistently outperforms all baseline models across all dimensions. Notably, general-purpose LLMs exhibit particularly weak performance in causality analysis, where TS-Reasoner achieves an 87% improvement over the best-performing baseline. This significant advantage is largely attributed to TS-Reasoner 's integration with program-aided tools for statistical analysis, enabling it to determine autocorrelation, lag relationships, and causal dependencies through numerical computations. The results underscore the benefits of task decomposition and external program utilization in enhancing time series understanding rather than plain text reasoning.

## 5.2 Multi-Step Time Series Inference

Table 2: Performance on Multi-Step Predictive Tasks. TSR denotes TS-Reasoner, TSR w/ PQ denotes TS-Reasoner prompted with paraphrased questions. LLM models are equipped with a CodeAct agent to receive execution feedback. The average MAPE values are calculated among all succeeded questions and the corresponding standard deviation is calculated.

| Constraint Type | Max Load | | Min Load | | Load Ramp Rate | | Load Variability | | Average | |
|---|---|---|---|---|---|---|---|---|---|---|
| | Success Rate | MAPE (Std) ↓ | Success Rate | MAPE (Std)↓ | Success Rate | MAPE (Std)↓ | Success Rate | MAPE (Std)↓ | Success Rate | MAPE ↓ |
| **Prediction w/ Covariates** | | | | | | | | | | |
| TSR | 1 | 0.0621 (0.10) | 1 | 0.0564 (0.06) | 1 | 0.0719 (0.20) | 0.9444 | 0.0577 (0.06) | **0.9861** | 0.0620 |
| TSR w/ PQ | 0.9474 | 0.0467 (0.08) | 1 | 0.0564 (0.06) | 0.9444 | 0.0659 (0.17) | 0.9444 | 0.0577 (0.06) | 0.9591 | **0.0567** |
| 4o | 1 | 0.0685 (0.10) | 1 | 0.1289 (0.19) | 0.7778 | 0.0847 (0.19) | 0.5 | 0.0443 (0.05) | 0.8194 | 0.0816 |
| o1 | 1 | 0.1289 (0.19) | 0.9 | 0.0748 (0.07) | 1 | 0.0861 (0.16) | 0.6667 | 0.0556 (0.06) | 0.8917 | 0.0864 |
| Deepseek | 1 | 0.1444 (0.22) | 1 | 0.1387 (0.16) | 0.8333 | 0.1180 (0.24) | 0.6667 | 0.0835 (0.07) | 0.8750 | 0.1212 |
| ReAct | 0.9474 | 0.0689 (0.10) | 0.9 | 0.0845 (0.09) | 0.8333 | 0.1021 (0.23) | 0.9444 | 0.0630 (0.07) | 0.9063 | 0.0796 |
| Chattime | 0.8421 | 0.1275(0.21) | 0.9 | 0.1158(0.14) | 0.6667 | 0.2522(0.34) | 0.7222 | 0.1725(0.18) | 0.7828 | 0.1670 |
| Chronos | 0.8947 | 0.1502(0.21) | 0.85 | 0.1205(0.22) | 0.4444 | 0.2153(0.31) | 0.7778 | 0.1223(0.15) | 0.7417 | 0.1521 |
| ARIMA | 0.7 | 0.2237(0.31) | 0.8 | 0.1579(0.22) | 0.85 | 0.1945(0.25) | 0.85 | 0.2805(0.15) | 0.8000 | 0.2142 |
| AutoGluon | 0.7368 | 0.0654 (0.23) | 0.8 | 0.1353(0.22) | 0.6111 | 0.0628(0.06) | 0.3888 | 0.0825(0.08) | 0.6342 | 0.0865 |
| **Prediction w/o Covariates** | | | | | | | | | | |
| TSR | 1 | 0.0799 (0.11) | 1 | 0.1366 (0.20) | 0.85 | 0.1191 (0.17) | 0.85 | 0.0767 (0.04) | **0.9250** | 0.1031 |
| TSR w/ PQ | 1 | 0.0806 (0.11) | 1 | 0.1366 (0.20) | 0.85 | 0.0763 (0.03) | 0.85 | 0.0812 (0.05) | **0.9250** | **0.0937** |
| 4o | 0.8 | 0.1607 (0.17) | 0.9 | 0.2007 (0.20) | 0.5 | 0.3430 (0.28) | 0.75 | 0.1838 (0.12) | 0.7375 | 0.2220 |
| o1 | 0.85 | 0.1518 (0.06) | 0.8 | 0.1439 (0.06) | 0.65 | 0.1433 (0.05) | 0.5 | 0.1672 (0.07) | 0.7000 | 0.1268 |
| Deepseek | 1 | 0.1612 (0.14) | 0.9 | 0.1978 (0.19) | 0.85 | 0.2042 (0.09) | 0.6 | 0.1655 (0.07) | 0.8375 | 0.1822 |
| ReAct | 0.55 | 0.1838 (0.12) | 0.65 | 0.2543 (0.17) | 0.4 | 0.4262 (0.37) | 0.75 | 0.2861 (0.22) | 0.5875 | 0.2876 |
| Chattime | 0.95 | 0.1207(0.17) | 0.75 | 0.1151(0.17) | 0.7 | 0.1142(0.07) | 0.8 | 0.0929(0.04) | 0.8000 | 0.1107 |
| Chronos | 0.9 | 0.1579(0.15) | 0.9 | 0.2142(0.24) | 0.8 | 0.2169(0.16) | 0.8 | 0.1543(0.05) | 0.85 | 0.1858 |
| ARIMA | 1 | 0.1457(0.14) | 0.95 | 0.1741(0.21) | 0.9 | 0.1616(0.11) | 1 | 0.1269(0.04) | 0.9625 | 0.1521 |
| AutoGluon | 0.6 | 0.1962(0.20) | 0.65 | 0.1335(0.06) | 0.55 | 0.1552(0.06) | 0.6 | 0.1471(0.04) | 0.6 | 0.1580 |
| **Prediction across Multiple Grids** | | | | | | | | | | |
| TSR | 0.9 | 0.1491 (0.18) | 1 | 0.1614 (0.18) | 0.9 | 0.1285 (0.13) | 1 | 0.1752 (0.25) | **0.9500** | 0.1535 |
| TSR w/ PQ | 0.85 | 0.1468 (0.18) | 0.95 | 0.1647 (0.18) | 0.95 | 0.1301 (0.12) | 0.9 | 0.1453 (0.19) | 0.9125 | **0.1467** |
| 4o | 0.8 | 0.2318 (0.30) | 0.85 | 0.3071 (0.35) | 0.6 | 0.3077 (0.36) | 0.55 | 0.5538 (0.44) | 0.7000 | 0.3501 |
| o1 | 0.85 | 0.3095 (0.37) | 0.85 | 0.2604 (0.26) | 0.95 | 0.1877 (0.23) | 0.7 | 0.3222 (0.34) | 0.8375 | 0.2708 |
| Deepseek | 0.85 | 0.1889 (0.25) | 1 | 0.1792 (0.23) | 0.75 | 0.3830 (0.40) | 0.55 | 0.9093 (0.29) | 0.7875 | 0.3416 |
| ReAct | 0.75 | 0.2229 (0.22) | 0.65 | 0.2110 (0.23) | 0.85 | 0.2591 (0.29) | 0.8 | 0.1790 (0.23) | 0.7625 | 0.2175 |
| Chattime | 0.8 | 0.1309(0.13) | 0.85 | 0.1411(0.14) | 0.6 | 0.1687(0.08) | 0.5 | 0.2039(0.25) | 0.6875 | 0.1612 |
| Chronos | 0.85 | 0.1324(0.19) | 0.8 | 0.1031(0.12) | 0.65 | 0.1280(0.07) | 0.65 | 0.1930(0.22) | 0.7375 | 0.1391 |
| ARIMA | 0.55 | 0.3975(0.34) | 0.6 | 0.4087(0.34) | 0.75 | 0.5100(0.31) | 0.85 | 0.3599(0.28) | 0.6875 | 0.4190 |
| AutoGluon | 0.85 | 0.0937(0.13) | 0.85 | 0.1386(0.22) | 0.75 | 0.0834(0.06) | | | | |

Given the novel paradigm of this task, there is a lack of suitable baselines models. For general purpose LLM agents, we adopt a code-based approach due to two primary challenges: loss of precision of time series data

---

[7]It was agreed by the original authors upon communication on this alternative metric that it better reflects the capability of LLMs.

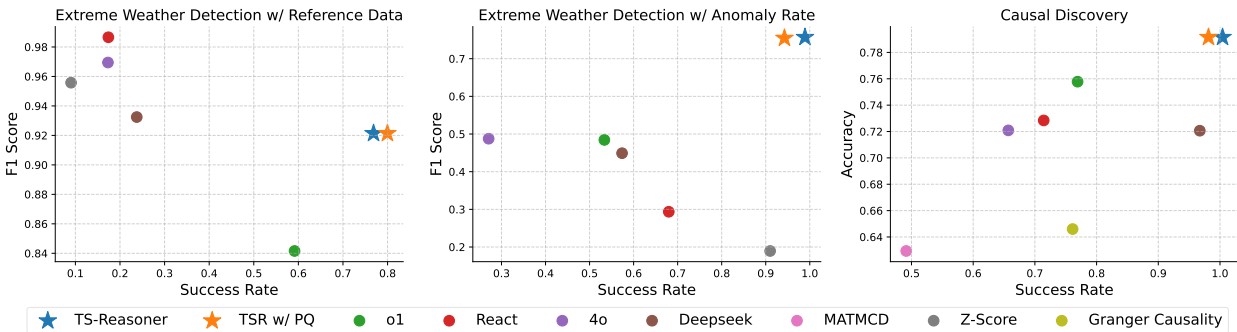

Figure 5: Performance on Multi-Step Diagnostic Tasks. A small jittering noise of 0.01 is added to success rate to distinguish overlapping points. TSR w/ PQ denotes TS-Reasoner when prompted with paraphrased questions.

in textual and image form and the limitation of context length encountered for tasks involving large amounts of time series data. To address these issues, we use the CodeAct agent implemented in agentscope (Gao et al., 2024; Wang et al., 2024a). The CodeAct agent is equipped with Jupyter Python interpreter, error feedback mechanism and standard machine learning and data processing libraries (e.g., statsmodels, scikit-learn, Granger causality utilities). Baseline models are prompted to autonomously generate code pipelines to solve the given tasks. Each agent has a maximum of $k$ interaction turns to reach a solution. In addition, we include ChatTime (Wang et al., 2025) and MATMCD (Shen et al., 2024) as representative multimodal time series reasoning models and include AutoGluon(Shchur et al., 2023) as an AutoML style baseline. To provide further reference points, we also evaluate classical statistical baselines, including ARIMA (Box & Pierce, 1970) for forecasting, Granger causality (Seth, 2007) for causal analysis, and Z-score–based methods (Rousseeuw & Hubert, 2011) for anomaly detection.

Table 2 presents the performance of TS-Reasoner and baseline models on predictive tasks. Mean Absolute Percentage Error (MAPE) is chosen as the primary inference quality metric to ensure scale-invariant evaluation. TS-Reasoner outperforms baseline models in success rates while simultaneously achieving superior inference quality as evidenced by lower MAPE values. MAPE values exceeding 1 are considered unreasonable inference and counted towards failures. Among successful task completions, TS-Reasoner demonstrates the most performance gain on prediction tasks without explicitly provided covariates tasks. This improvement is largely attributed to its ability to retrieve relevant covariates deemed necessary by the task decomposer.

Figure 5 shows the performance of TS-Reasoner and baseline models for anomaly detection with threshold calibration and causal discovery with domain knowledge tasks. Methods positioned toward the upper-right direction which corresponds to higher success rates and higher inference quality among successful cases are preferred. While some baselines attain higher inference-quality scores on a small subset of successful executions, they exhibit substantially lower success rates. In contrast, TS-Reasoner achieves the strongest overall tradeoff, consistently attaining high success rates together with competitive inference quality. This highlights the effectiveness of TS-Reasoner in orchestrating structured, operator-based workflows for complex multi-step time series inference, rather than relying on isolated model performance.

**Error Analysis** In this section, we examine the failure modes of TS-Reasoner and baseline models. By systematically categorizing errors into execution failures, constraint violations, and insufficient/trivial predictions, we can isolate the fundamental weaknesses in current approaches. Execution failures include runtime errors or invalid outputs, while constraint violations refer to outputs that ignore task-specific requirements such as anomaly rates or variability limits. Inadequate results are defined as those yielding poor utility, such as MAPE > 1, forecasting accuracy = 0, or F1 score = 0. Figure 6 presents error distributions across different models on the electricity load prediction task without covariates. TS-Reasoner achieves a stable success rate under both the original and paraphrased-instruction settings, indicating robustness to instruction variation. The remaining failures are dominated by constraint violations (6.2–7.5%), with the paraphrased setting exhibiting a slightly higher rate, which we attribute to increased ambiguity and diversity in how constraints

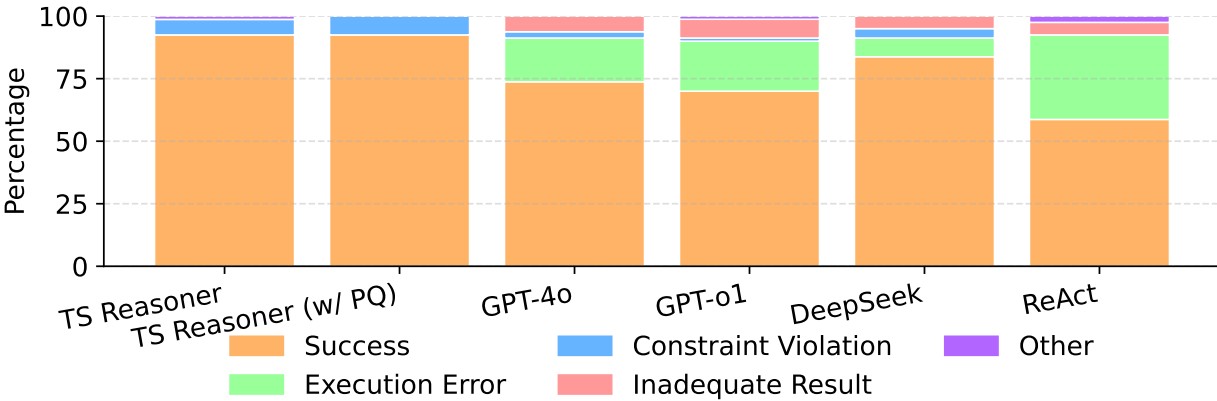

Figure 6: Error distribution of different approaches on electricity prediction task without covariates.

Table 3: Performance Comparison with key operators removed on constrained forecasting task.

| Constraint Types | TS-Reasoner | | No Custom Operator | | No Retrieval Operator | |
|---|---|---|---|---|---|---|
| | SR | MAPE ↓ | SR | MAPE ↓ | SR | MAPE ↓ |
| **Max Load** | 1.00 | 0.0799 | 0.85 | 0.0557 | 1.00 | 0.1349 |
| **Min Load** | 1.00 | 0.1366 | 0.80 | 0.1559 | 1.00 | 0.1723 |
| **Load Ramp Rate** | 0.85 | 0.1191 | 0.15 | 0.1193 | 0.95 | 0.1707 |
| **Load Variability** | 0.85 | 0.0767 | 0.80 | 0.0843 | 1.00 | 0.1233 |
| **Average** | 0.925 | 0.1031 | 0.65 | 0.1038 | 0.9875 | 0.1503 |

are expressed rather than execution instability. In contrast, baseline models display more frequent execution failures and trivial predictions, leading to substantially lower overall reliability.

General-purpose LLMs struggle primarily with execution errors, indicating difficulties generating valid computational pipelines for time series analysis. The ReAct agent, despite its explicit reasoning approach, achieves only a 58.7% success rate with the highest proportion of execution errors (33.8%) due to the complete lack of error feedback. The absence of execution errors in TS-Reasoner highlights the effectiveness of its modular architecture with well-tested computational components. This analysis underscores the necessity of domain-specialized agents like TS-Reasoner for complex time series tasks in real-world applications, as evidenced by its 19-34% higher success rates compared to baseline approaches. On the other hand, ReAct agent with explicit thinking can significantly reduce the constraint violation rate but still suffers from execution failures and generating trivial predictions.

**Ablation Study** Figure 7 presents an ablation study on the Electricity Prediction with Covariates task, where we progressively remove key components of TS-Reasoner to assess their individual contributions. The full agent achieves a 92.5% success rate with a MAPE of 0.1031. Removing intermediate feedback slightly reduces the success rate to 91.3%, with inadequate results beginning to emerge. When both intermediate and error feedback are removed, the success rate drops further to 90.0%, and execution errors begin to appear. The most significant degradation occurs when special operators are removed, reverting to a general-purpose LLM executor equipped with CodeAct. In this condition, the success rate falls sharply to 73.8%, and the MAPE more than doubles to 0.2220. These results demonstrate that feedback mechanisms play a critical role in securing a high success rate, while specialized time series operators are essential for maintaining strong inference quality.

We further isolate the effects of key operator categories in Table 3. Removing the LLM-generated custom operator substantially reduces the success rate (from 92.5% to 65.0% on average), as the agent loses the

Table 4: Causal Knowledge task: generalization to more ambiguous qualitative domain knowledge with a single added in-context example. Stock Price Future Volatility task: generalization to unseen domain and instruction.

| Task | Causal Known Prior | | Stock Future Volatility | |
|---|---|---|---|---|
| | SR | Acc ↑ | SR | MAPE ↓ |
| **TSR** | **1** | **0.739 (0.13)** | **0.98** | **0.620 (0.23)** |
| **4o*** | 0.68 | 0.674 (0.16) | 0.28 | 0.837 (0.19) |
| **o1*** | 0.7 | 0.674 (0.14) | 0.42 | 0.841 (0.18) |
| **DS*** | 0.76 | 0.682 (0.15) | 0.2 | 0.745 (0.30) |
| **ReAct** | 0.66 | 0.659 (0.15) | 0.16 | 0.970 (0.06) |

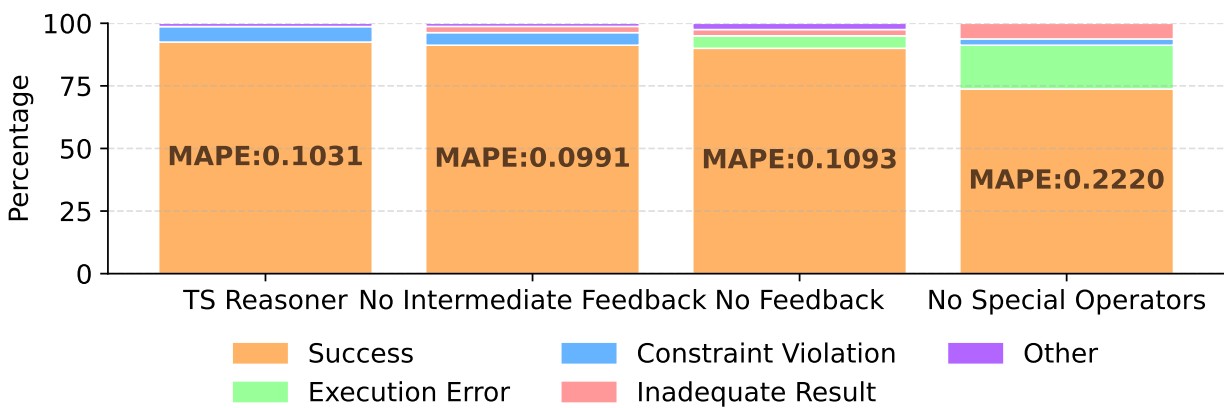

Figure 7: Ablation Study on Electricity Prediction w/ Covariates task. We first remove intermediate feedback only, then we remove both intermediate and error feedback (denoted as No Feedback), and lastly we remove the special operators and falling back to the general purpose LLM equipped with CodeAct.

ability to perform task-specific refinement under constraints. In contrast, removing external data retrieval operators preserves a high success rate but leads to a pronounced degradation in inference quality, with MAPE increasing by approximately 50% on average. This divergence highlights a clear functional separation: custom operators are critical for ensuring successful task completion under constraints, whereas retrieval operators play a key role in enriching contextual information and improving prediction accuracy.

## 6 Analysis

**Generalizing to Qualitative Constraints and New Domains** We further consider the practical utility of TS-Reasoner to handle novel scenarios without extensive retraining or specialized adaptation. We evaluate TS-Reasoner's adaptability to novel constraint types and previously unseen domains through two generalization tasks, with results presented in Table 4. First, we introduce a causal discovery task incorporating qualitative constraints that require reasoning about directional relationships rather than precise numerical quantity. With just a single additional in-context example added to demonstrate the handling of such qualitative constraints, TS-Reasoner achieves strong performance over baselines, demonstrating effective transfer learning capabilities. Secondly, we test adaptation to stock future volatility prediction task featuring unseen domain and instruction from the in-context samples. TS-Reasoner requires no explicit in-context examples for this task; instead, the task decomposer successfully generalizes by using its knowledge of toolkit functionalities to autonomously construct valid solution paths. These results demonstrate TS-Reasoner's generalization capabilities across both new constraint and application domains.

## 7  Broader Impact and Limitation

Compared to traditional time series models, TS-Reasoner extends inference capabilities by supporting compositional reasoning and multi-step execution. However, it is intended as a decision-support and workflow prototyping framework, rather than a fully autonomous time series modeling solution. TS-Reasoner only represents an initial step toward multi-step time series inference and cannot fully understands temporal signals. Unlike multimodal foundation models, which encode time series data, TS-Reasoner does not process raw time series data directly. Instead, it relies on task decomposition and structured execution. Further research effort is still needed to address challenges of semantic space alignment between time series and natural language. One limitation of TS-Reasoneris that generalizing to new tasks that requires tools beyond current toolbox may require additional toolbox engineering such as visualization tools. In addition, the success of TS-Reasoner relies in part on manually annotated in-context examples that demonstrate proper task decomposition and tool usage. A future direction of this work is to adaptively refine the in-context set by learning from execution errors and feedback signals, enabling the model to iteratively improve its reasoning workflows over time.

## 8  Conclusion

In this study, we introduced TS-Reasoner, a time series domain-specialized time series agent for multi-step inference tasks. TS-Reasoner integrates compositional reasoning with structured execution of expert tools. Through comprehensive benchmarking on TimeSeriesExam and a diverse set of real-world-inspired tasks that demonstrate multi-step complexity in nature, we evaluated TS-Reasoner's performance against leading general-purpose LLM-based agents. Our experiments demonstrate that TS-Reasoner consistently outperforms baseline models in both success rate and inference quality. The results reveal limitations of current general-purpose LLMs in multi-step time series analysis tasks, including frequent execution errors and suboptimal outputs. These findings suggest that LLMs alone are insufficient for such tasks, and that a hybrid approach combining LLMs with domain-specific operators is essential. By bridging LLM reasoning with time series analytical workflows, TS-Reasoner emerges as a simple yet effective solution, paving the way for more intelligent time series agents.

## 9  Acknowledgment

This work was supported in part by the Department of Defense under Cooperative Agreement Number W911NF-24-2-0133 as well as NSF project # 2425919 and #2413417. The views and conclusions contained in this document are those of the authors and should not be interpreted as representing the official policies, either expressed or implied, of the Army Research Office or the U.S. Government. The U.S. Government is authorized to reproduce and distribute reprints for Government purposes notwithstanding any copyright notation herein.

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

| Hist | Pred | Dataset-Specific Forecast | | | | Dataset-Shared Forecast | | | | |
|---|---|---|---|---|---|---|---|---|---|---|
| | | DLinear | GPT4TS | TimeLLM | TGForecaster | Moirai | TimesFM | Chronos | ChatTime | TS-Reasoner |
| 48 | 24 | 0.5204 | 0.4373 | 0.4211 | 0.4411 | 0.5981 | 0.4851 | 0.4813 | 0.4849 | **0.4194** |
| 72 | 24 | 0.5075 | 0.4253 | 0.4031 | 0.3943 | 0.5776 | 0.4258 | 0.4276 | 0.4307 | **0.3619** |
| 96 | 24 | 0.4965 | 0.3921 | 0.4392 | 0.3653 | 0.5179 | 0.4054 | 0.4336 | 0.3920 | **0.3538** |
| 120 | 24 | 0.4796 | 0.3713 | 0.3594 | 0.3594 | 0.5245 | 0.3807 | 0.3902 | 0.3480 | **0.3336** |
| Avg. MAE | | 0.5010 | 0.4065 | 0.4057 | 0.3900 | 0.5545 | 0.4243 | 0.4332 | 0.4139 | **0.3672** |

Table 5: Evaluation results on the PTF context-guided time series forecasting task. Lower values indicate better performance.

| Feat | Len | GPT4 | GPT3.5 | GLM4 | LLaMA3 | ChatTime | TS-Reasoner |
|---|---|---|---|---|---|---|---|
| Trend | 64 | 0.6532 | 0.3507 | 0.7319 | 0.6799 | 0.9011 | **0.9967** |
| | 128 | 0.7015 | 0.5846 | 0.7574 | 0.5855 | 0.9068 | **0.9927** |
| | 256 | 0.7482 | 0.5028 | 0.6377 | 0.6143 | 0.8843 | **0.9687** |
| | 512 | 0.6346 | 0.5903 | 0.6697 | 0.6753 | 0.8234 | **0.9343** |
| Volatility | 64 | 0.5585 | 0.5633 | 0.6797 | 0.6373 | 0.7874 | **0.8900** |
| | 128 | 0.4979 | 0.3839 | 0.4770 | 0.4756 | 0.6954 | **0.9803** |
| | 256 | 0.4624 | 0.4894 | 0.5418 | 0.5246 | 0.6228 | **0.9713** |
| | 512 | 0.3169 | 0.3796 | 0.4549 | 0.5261 | 0.5736 | **0.6493** |
| Season | 64 | 0.3518 | 0.3428 | 0.3366 | 0.3484 | 0.6639 | **0.8627** |
| | 128 | 0.3515 | 0.3952 | 0.3464 | 0.3958 | 0.6517 | **0.8767** |
| | 256 | 0.5283 | 0.5089 | 0.3892 | 0.4120 | 0.6463 | **0.8810** |
| | 512 | 0.4457 | 0.4889 | 0.3892 | 0.4127 | 0.6244 | **0.8890** |
| Outlier | 64 | 0.7230 | 0.4325 | 0.5359 | 0.7051 | 0.8773 | **0.8787** |
| | 128 | 0.6327 | 0.5940 | 0.5298 | 0.5694 | 0.9032 | **0.9827** |
| | 256 | 0.6795 | 0.4579 | 0.5019 | 0.5073 | 0.8593 | **0.9913** |
| | 512 | 0.6219 | 0.4996 | 0.2822 | 0.4085 | 0.7478 | **0.9910** |
| Avg. Acc | | 0.5567 | 0.4728 | 0.5163 | 0.5299 | 0.7605 | **0.9210** |

Table 6: The evaluation result in the time series question answering task. Higher values mean better performance for all metrics. The best results are highlighted in bold.

# A    Additional Evaluation

Table 6 presents results on the TSQA benchmark, which evaluates question answering accuracy across feature groups (trend, volatility, seasonality, and outliers) and increasing sequence lengths (64/128/256/512). A notable difference between the two methods is their behavior as sequence length increases. ChatTime's performance generally degrades with longer sequences across most feature groups, whereas TS-Reasoner maintains stable accuracy as sequence length grows with the exception of Volatility. This suggests that TS-Reasoner is less constrained by sequence length due to its operator-based execution and decomposition strategy, rather than relying on direct sequence encoding within the language model. We further evaluate TS-Reasoner on established benchmarks from the ChatTime framework to assess performance beyond our proposed multi-step inference tasks. Table 5 reports results on the PTF dataset for context-guided forecasting. TS-Reasoner achieves lower MAE than ChatTime across all evaluated history lengths.

# B    Reliability of Base Model & Cost

**Base Model Reliability**    To assess the robustness of TS-Reasoner with respect to the choice of base language model, we substitute ChatGPT-4o (Hurst et al., 2024) with a smaller open-source backbone, LLaMA 3.1 70B (Dubey et al., 2024), and evaluate performance across the same set of tasks (Table 2). Despite the reduced model capacity, TS-Reasoner maintains comparable task success rates and inference

quality across all evaluated settings. This result indicates that TS-Reasoner's effectiveness does not rely solely on the raw reasoning capacity of the underlying language model; instead, its structured execution pipeline and modular operator design enable robust performance even with weaker decomposer models. These findings suggest that TS-Reasoner can be deployed reliably across environments with varying LLM resources.

**Inference Cost**   Given this backbone-agnostic behavior, we next analyze the inference cost of TS-Reasoner under a multi-turn execution setting. We estimate cost by explicitly separating *new input tokens*, *cached input tokens*, and *output tokens*. On average, each turn produces 63 output tokens, priced at $10 per million tokens. With a maximum of 5 interaction turns, this results in an output cost of

$$63 \times 5 \times 10/1{,}000{,}000 \approx \$0.003.$$

The initial input prompt contains 2,647 tokens and is billed at the new-token rate of $2.5 per million tokens. In subsequent turns, previously seen inputs are reused and charged at the cached-token rate of $1.25 per million tokens. Accounting for token reuse across turns, the cumulative cached-input cost is

$$2647 \times (5 + 4 + 3 + 2) \times 1.25/1{,}000{,}000 \approx \$0.49.$$

Overall, the total inference cost is approximately $0.5 per question. Consistent with the reliability analysis above, reliance on proprietary models is not essential: when using an open-source backbone (LLaMA 3.1 70B), TS-Reasonerachieves comparable task success rates (94.83% vs. 94.25%) and similar inference quality, while avoiding proprietary API usage.

Table 7: Performance Comparison between TS-Reasoner (GPT-4o) and TS-Reasoner (LLaMA 3.1 70B) across various tasks.

| Task Decomposer | GPT-4o | | | LLaMA 3.1 70B | | |
| Task | Success Rate | MAPE | Std | Success Rate | MAPE | Std |
|---|---|---|---|---|---|---|
| **Electricity w/ Covariates** | | | | | | |
| Max Load | 1 | 0.0799 | 0.1128 | 1 | 0.0990 | 0.1055 |
| Min Load | 1 | 0.1366 | 0.1951 | 1 | 0.1623 | 0.2169 |
| Load Ramp Rate | 0.85 | 0.1191 | 0.1663 | 0.85 | 0.0928 | 0.0363 |
| Load Variability | 0.85 | 0.0767 | 0.0422 | 0.95 | 0.1534 | 0.2028 |
| **Electricity w/ Covariates** | | | | | | |
| Max Load | 1 | 0.0621 | 0.1037 | 1 | 0.0677 | 0.1020 |
| Min Load | 1 | 0.0564 | 0.0605 | 1 | 0.0615 | 0.0539 |
| Load Ramp Rate | 1 | 0.0719 | 0.1991 | 1 | 0.0813 | 0.2032 |
| Load Variability | 0.9444 | 0.0577 | 0.0588 | 0.8889 | 0.0692 | 0.0928 |
| **Prediction Across Multiple Grids** | | | | | | |
| Max Load | 0.9 | 0.1491 | 0.1750 | 0.85 | 0.1503 | 0.1800 |
| Min Load | 1 | 0.1614 | 0.1773 | 1 | 0.1614 | 0.1773 |
| Load Ramp Rate | 0.9 | 0.1285 | 0.1269 | 0.9 | 0.1324 | 0.1257 |
| Load Variability | 1 | 0.1752 | 0.2455 | 0.9 | 0.1938 | 0.2520 |
| **Task** | **Success Rate** | **F1** | **Std** | **Success Rate** | **F1** | **Std** |
| **Extreme Weather Detection w/ Reference Data** | 0.78 | 0.9214 | 0.1082 | 0.8 | 0.9232 | 0.1074 |
| **Extreme Weather Detection w/ Anomaly Rate** | 1 | 0.7569 | 0.0556 | 1 | 0.7569 | 0.0556 |
| **Task** | **Success Rate** | **Accuracy** | **Std** | **Success Rate** | **Accuracy** | **Std** |
| **Causal Discovery** | 1 | 0.7915 | 0.1246 | 1 | 0.7909 | 0.1239 |

# C   Operator Description

**Time Series Models operators**
- UniPreOP: This operator is primarily used for univariate time series forecasting, leveraging advanced time series models to accurately predict future sequences. It currently supports Lag-Llama, Chronos, TimeGPT, TEMPO, Arima zero shot inference.

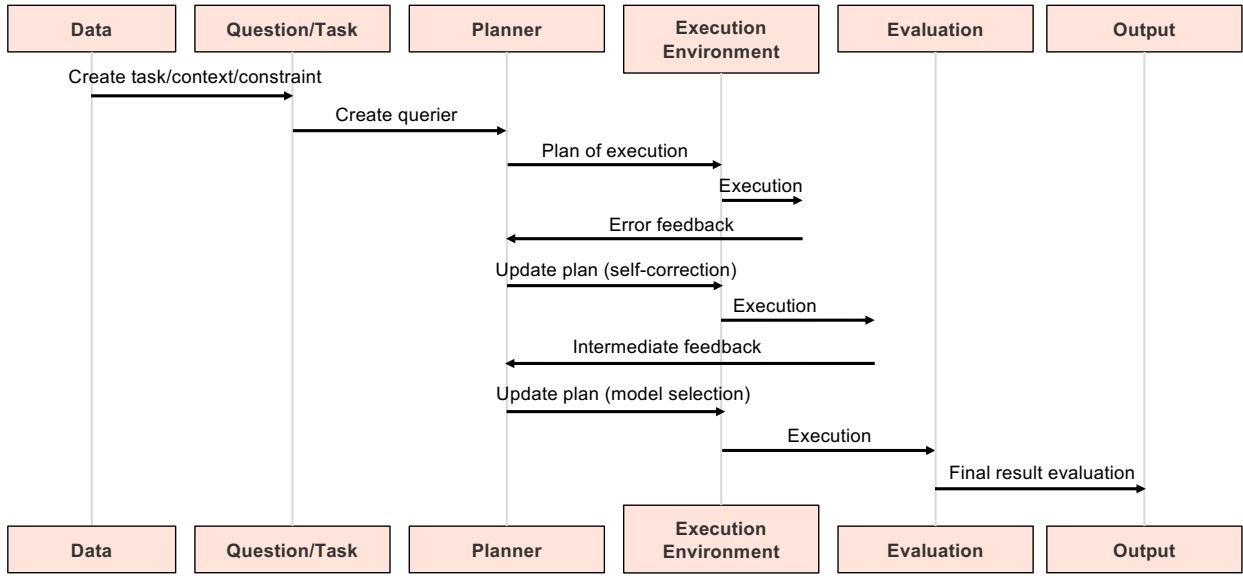

Figure 8: Illustration of an example TS-Reasoner workflow

| Error Type | Inadequate Result | Constraint Violation | Others (Shape Misalignment) |
|---|---|---|---|
| **User Instruction** | I have 2m_temperature data that spans 279 hours. Please tell me whether there are anomalies (extreme weather events) and where are anomalies if present in this sequence. Please return a 1D numpy array with 1 indicating an anomaly and 0 indicating no anomaly. | I have historical Wind Speed, Relative Humidity data and the corresponding wind_power data for the past 167 minutes. I need to manage the load variability so that it does not exceed 0.0305 MW over the given period. Think about how Wind Speed, Relative Humidity influence wind_power. Please give me a forecast for the next 65 minutes for wind_power. Your goal is to make the most accurate forecast as possible, refine prediction result based on the constraint previously described. | I have historical Dew Point, Relative Humidity, Solar Zenith Angle data and the corresponding load_power data for the past 108 minutes. I require that the system load is maintained above a minimum of 0.70 MW. Think about how Dew Point, Relative Humidity, Solar Zenith Angle influence load_power. Please give me a forecast for the next 68 minutes for load_power. |
| **Output** | Final_value: [0,0,0,1,1,0,0,0,…] | Final_value: [ 0. ... 0.03612244 ...] | Final_value: [0.87 0.89 ... 0.88] Final_value.shape = (67,) |
| **Evaluation Result** | {'status':0, 'message': 'unreasonable prediction, f1 score is 0.0'} | {'status': 0, 'message': 'Predicted load variability exceeds the maximum allowable limit of 0.0305 MW. | {'status':0, 'message': 'Prediction Shape Misalignment, Expecting output of length 68', 'error':1} |

Figure 9: Example Result Errors

- MultiPreOP: This is a multivariate time series forecasting operator used to accurately predict the target variable based on multiple exogenous variables. It currently support TimeGPT zero shot inference and linear regression.

- AnomalDetOP: This is a time series anomaly detection operator used to generate anomaly scores based on reconstruction errors. It current support MOMENT foundation model zero shot inference.

- trainForecastOP: This is a operator that supports training time series forecasting models given input data. It currently supports iTransformer training. Model choice is based on current Time Series Library Leaderboard and more models can be included.

- trainADOP: This is a operator that supports training time series anomaly detection models given input data. It currently supports TimesNet training. Model choice is based on current Time Series Library Leaderboard and more models can be included.

**Numerical operators**
- CausalMatrixOP: This operator is used to calculate significance of Granger causality relationship between each pair of variables in a time series dataset.

- ConcatOP: This operator concatenates the two given data horizontally.

- thToBinaryOP: This operator converts the input data into binary values based on threshold or percentile value.

- calibrateThreshOP: This operator approximates the input data with a normal distribution and returns the critical threshold value (3 standard deviations away from the mean) of this distribution.

- ApplyOP: This operator applies input function to the corresponding data. There are many more operators in this category that performs tasks as suggested by their names: checkStationaryOP (perform adf test), checkTrendStationaryOP (perform kpss test), getChptOP (perform bayesian changepoint detection), getTrendOP, compareDisOP (perform ks test), detectSpikesOP, getCyclePatternOP, getAmplitudeOP, getPeriodOP, testWhiteNoiseOP (perform box test), getNoiseCompOP, getAutoCorrOP, getMaxCorrLagOP, decomposeOP, getSlidingVarOP, getCyclePatternOP, detectFlippedOP, detectSpeedUpDownOP, detectCutoffOP, getTrendCoefOP, VolDetOP

**Data retrieval operators**
- getEnvDataOP: This operator can retrieve specified weather or air quality variables if available from open-meteo API. The input must specify a time range, geolocation, as well as resolution (daily or hourly).

-getElectricityDataOP: This operator retrieves electricity data given zone code from the official EIA website. The input must specify a grid zone name, time range, and variables to retrieve.

**LLM generated custom operators**
- RefGenOP: This operator is primarily used to generate a corresponding python function based on the requirement described in the prompt.

# D   Error Examples and Additional Error Analysis

This section illustrates an example execution workflow of TS-Reasoner (figure 8) and representative failure cases, including execution errors, shape mismatches, constraint violations, and inadequate predictions (figures 9 and 14). Examples are drawn from both TS-Reasoner and baseline models. Figures 15, 10, 11, 12, and 13 show the distribution of error types across various task settings. Execution errors in TS-Reasoner are minimal, reflecting the effectiveness of operator-based abstraction. However, introducing paraphrased questions occasionally triggers execution failures, suggesting sensitivity to linguistic variation. In extreme weather detection task (figure 11), inadequate result is the most dominant failure patterns, demonstrating an inability of baseline models to fully leverage anomaly free reference samples to calibrate anomaly threshold often producing a trivial all-zero labels.

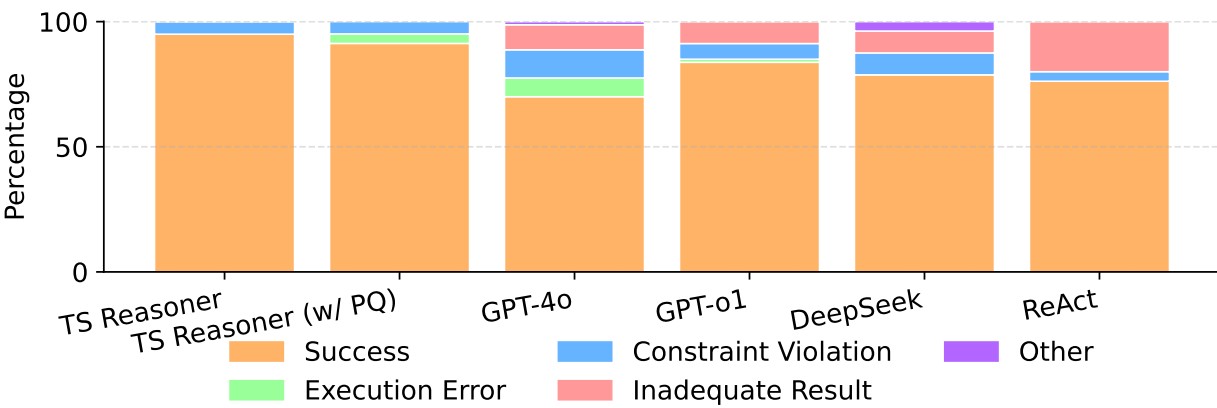

Figure 10: Error distribution of different approaches on electricity prediction task across multiple grids.

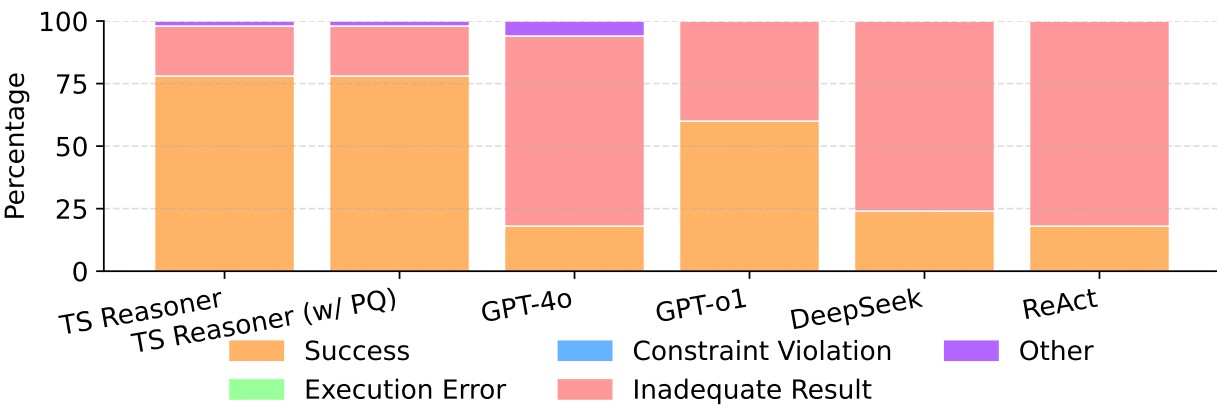

Figure 11: Error distribution of different approaches on extreme weather detection with anomaly free reference data.

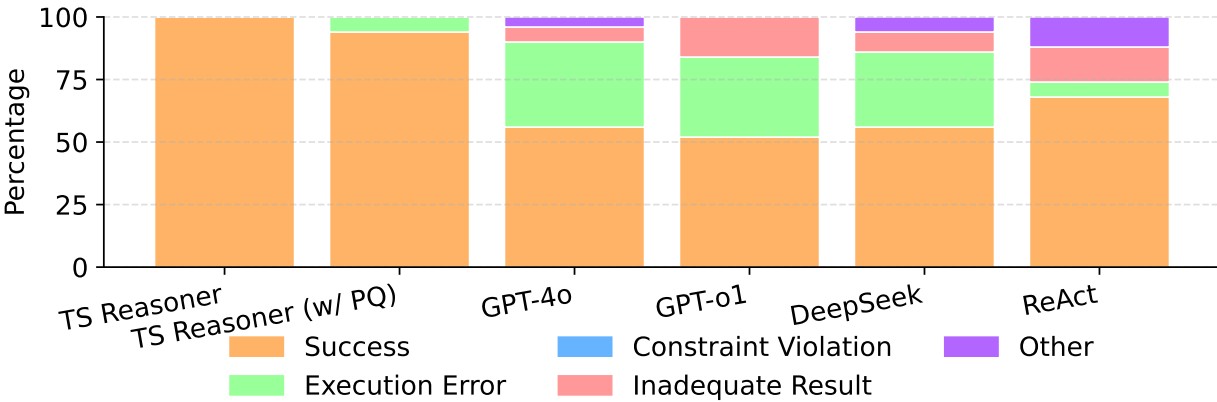

Figure 12: Error distribution of different approaches on extreme weather detection task with known anomaly rates.

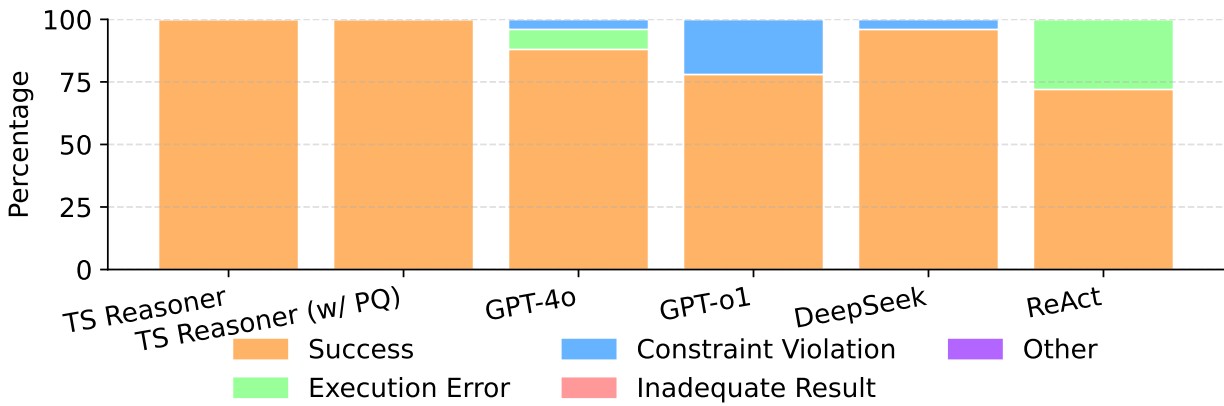

Figure 13: Error distribution of different approaches on causal discovery tasks.

| Error Type | Execution Error | Execution Error |
|---|---|---|
| User Instruction | I have historical Advertising Spend (A), Sales (B), Economic Factors (C), Customer Sentiment (D) data and want to get the causal relationship between each pair of the variables. I know that 41.66666666666667% of the variable pairs have relationship. Consider the potential influence of each variable on the others in this variable list: ['Advertising Spend (A)', 'Sales (B)', 'Economic Factors (C)', 'Customer Sentiment (D)']. Please provide 2d numpy matrix with binary values to indicate whether each pair of variables has a relationship. | I have load_power data for the past 135 minutes. I require that the system load is maintained above a minimum of 1490.6020757896545 MW.  Please give me a forecast for the next 17 minutes for load_power. Think about what could be relevant covariates that can help forecast load_power. Your goal is to make the most accurate forecast as possible, refine prediction result based on the constraint previously described, and return the result as a 1D numpy array. |
| Execution | An error occurred: name 'n_variables' is not defined | An error occurred: can't multiply sequence by non-int of type 'float' final value. |
| Evaluation Result | {'status':0, 'message': "evaluator received input of NoneType"} | {'status':0, 'message':"An error occurred: 'Index' object has no attribute 'minute'"} |

Figure 14: Example Execution Errors.

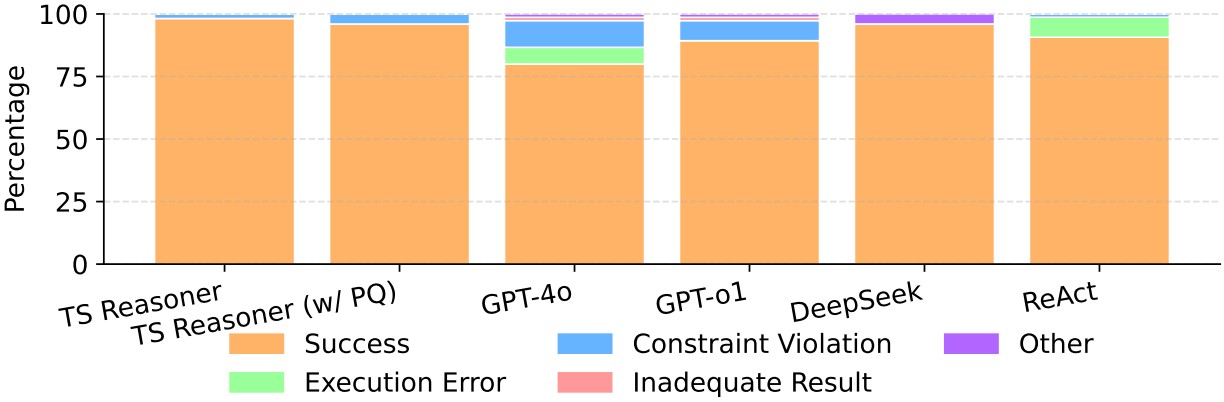

Figure 15: Error distribution of different approaches on electricity prediction task with covariates.

## E Prompt for TS-Reasoner

```
"""
def UniPreOP(data, future_length, model):
    Channel independent zero shot forecasting tool. For time series with
        covariates, this tool can be used by simply providing the main time series
        data.

    Input:

    - data: np.array([T,C]),T=history data len, C=variable number, the main time
        series

    - future_length: int, the number of steps to forecast

    - model: str, the model to use for prediction. available models:  "TEMPO", "
        ARIMA", "Chronos", "Moirai", "TabPFN", "Lagllama"

    Output:

    - np.array([N,C]), N=future\_length, C=variable number, the predicted future
        values

(more definitions of operators)

Example:

Question:
I have 2m_temperature data that spans 219 hours. Please tell me whether there are
    anomalies (extreme weather events) and where are anomalies if present in this
    sequence. Please return a 1D numpy array with 1 indicating an anomaly and 0
    indicating no anomaly. The data is stored in variable VAL and some anomaly-free
     normal samples are stored in variable NORM_VAL.
Program:
NORM_SCORE = AnomalDetOP(data=NORM_VAL)
THRES = calibrateThreshOP(data=NORM_SCORE)
TEST_SCORE = AnomalDetOP(data=VAL)
FINAL_RESULT = convertBinaryOP(data=TEST_SCORE, threshold=THRES)

(a few more examples)

Question:

I have 2m temperature data that spans 202 hours. Please tell me whether there are
    anomalies (extreme weather events) and where are anomalies if present in this
    sequence. Please return a 1D numpy array with 1 indicating an anomaly and 0
    indicating no anomaly. The data is stored in variable VAL and some anomaly-free
     normal samples are stored in variable NORM_VAL. Follow previous examples and
    answer my last question in the same format as previous examples within markdown
     format in ```python```.

Given the question and the toolbox definition provided above, please give steps of
     operations from the oprations availble listed above that can answer the
    question. Return only toolbox operations and enclose your response in ```python
    ``` markdown. Do not include any other information in your response. For *PreOP
     modules, you can recieve intermediate evaluation feedback to help you choose
    the best model. You should revert to a previously best performing model if all
    other models you tried fail to improve the performance (a lower MAPE value
    indicates better performance). You will be able to regenrate your response upon
```

```
    recieving feedback. Your goal is to achieve a status of 1 in the evaluation
   result when possible. For example, if model A gives lower MAPE value compared
   to model B, you should choose model A.You should replace {PLACEHOLDER} with the
    corresponding arguments you deem appropriate for the operation functions. DO
   NOT use any other irrelevant operations or custom python code, follow my
   examples where solutions should be in such format.
"""
```

| Q. No. | Question | Ambiguity Level (Reasons) |
|---|---|---|
| 1 | Detecting anomalies in a 2-meter temperature time series dataset that spans 8.72 days (824 hours) and consists of 253 data points, I need to identify extreme weather events and determine their locations within the sequence. This requires labeling a binary numpy array where 1 indicates an anomaly and 0 indicates a normal value. The input data is stored in the variable "VAL" and the expected anomaly rate is to be stored in the variable "ANOMALY_RATE". | Low (Clear input and output, well-defined task, specific variable names) |
| 2 | Indicate the presence of extreme weather events within a 2m_temperature dataset spanning 260 hours. Using the provided VAL and NORM_VAL arrays, provide a one-dimensional numpy array where each entry corresponds to one hour, tagged with 1 for an identified anomaly and 0 for a typical observation. | Medium (Did not explain what data each variable contains) |
| 3 | Given a 2D array of temperature data with a duration of 241 hours, I'd like to determine and visualize the occurrence of anomalous events and their corresponding positions within the sequence. To achieve this, I'd appreciate it if you could provide an output array of the same length as the original data, where each element indicates whether the corresponding temperature reading is anomalous (1) or not (0), alongside some sample normal data for reference. | Medium (Did not define what is anomalous event but it should be referring to extreme weather events, no clear input variables) |
| 4 | Given a dataset of system load power over the past 123 minutes, we need to develop a 69-minute forecast that adheres to a maximum allowable system load of 694.4796 MW. The goal is to leverage relevant historical data and make an accurate load power prediction, considering constraints and return the forecast as a one-dimensional numpy array. | Medium (Unclear where data is stored in, did not define relevant historical data when it refers to historical load data) |
| 5 | Provide a 26-minute load power forecast for 17 distinct electric grid zones, constrained by a maximum allowable system load of 0.9872 MW per zone, based on historical data (1030 minutes) with 17 associated load_power values. | High (No input variable names, unclear output format) |
| 6 | What historical wind power data for 19 electric grid zones over 882 minutes can be utilized to create a 56-minute wind power forecast while ensuring each zone's system load remains above a minimum threshold of approximately 0.01 MW. The goal is to achieve the most accurate forecast possible, taking into account the specified constraint. | High (Unclear input variable names, the constraint number is approximated) |
| 7 | Given 70 minutes of historical load_power data in variable VAL, derive a 37-minute forecast for future load_power while ensuring that the predicted maximum system load does not exceed 1151.4102 MW. What relevant covariates should be considered for this forecast, and how can the accuracy of the prediction be refined given this constraint to produce a 1D numpy array result? | High (Multiple tasks mixed, formulated as questions rather than task instructions) |

Table 8: Sample paraphrased questions classified based on ambiguity levels.

