# OpenReview forum: "TS-Reasoner: Domain-Oriented Time Series Inference Agents for Reasoning and Automated Analysis"
_TMLR — Accepted by TMLR_

### Review · Reviewer_axWm · 2025-12-26

**Summary Of Contributions:**

Summary:

This paper presents TS-Reasoner, a domain-oriented time series inference agent that augments large language models with structured operator-based workflows to enable multi-step, constraint-aware time series reasoning. By decomposing natural language queries into executable analytical pipelines and incorporating error feedback for self-refinement, the framework addresses key limitations of general-purpose LLMs in numerical precision and procedural reasoning. The authors further introduce a realistic benchmark covering predictive and diagnostic tasks with domain constraints, and demonstrate that TS-Reasoner consistently outperforms strong LLM-based baselines in both success rate and inference quality. Overall, the work highlights the promise of domain-specialized LLM agents for automating complex, real-world time series analysis.

Strengths:

1. The paper clearly identifies a practical gap in current time series modeling and LLM-based approaches, namely the lack of support for multi-step, constraint-aware inference.
2. The proposed TS-Reasoner architecture is well-motivated and modular, effectively separating high-level reasoning from numerical execution to ensure robustness and precision.
3. The introduction of a realistic, domain-grounded benchmark significantly strengthens the empirical evaluation and goes beyond standard single-task settings.
4. Experimental results are comprehensive and demonstrate consistent improvements over strong general-purpose LLM baselines across diverse time series tasks.

Weaknesses:

1. The proposed approach relies heavily on manually designed operator libraries and in-context examples, which limits the overall scalability and level of automation of the system.
2. TS-Reasoner does not directly model raw time series data; instead, it processes time series through external tools, which constrains its potential for deep integration with time series foundation models.
3. The stability and interpretability of the LLM acting as the task decomposer are not systematically analyzed, particularly regarding robustness under complex or ambiguous instructions.
4. The newly introduced dataset is relatively limited in scale; although the task types are diverse, it still falls short of fully capturing the complexity and noise present in real-world industrial scenarios.
5. The experimental evaluation primarily relies on a small number of high-performance proprietary or large backbone models, leaving the adaptability of the framework to smaller or resource-constrained models insufficiently explored.
6. From a research perspective, the work is more oriented toward time series systems and engineering-focused agent design, with relatively limited connection to current core research hotspots in large language models, such as model architectures, training paradigms, or general-purpose reasoning mechanisms.

**Audience:**

Yes

**Audience Explanation:**

Yes. The paper would be of interest to researchers working on time series analysis, LLM-based agents, and domain-specialized reasoning, as it addresses practical limitations of general-purpose LLMs and proposes a concrete framework and benchmark for multi-step, constraint-aware inference.

**Claims And Evidence:**

Yes

**Claims Explanation:**

Yes. The claims are supported by clear and consistent experimental evidence, including evaluations on existing benchmarks and a newly proposed dataset, comparisons with strong baselines, and ablation studies that validate the effectiveness of the proposed design.

**Requested Changes:**

See the Weaknesses.

---

> ### Author Response · Authors · 2026-01-20
>
> We thank the reviewer for the thorough and thoughtful evaluation. We appreciate the reviewer’s recognition of the clear motivation, modular design, as well as the consistent empirical improvements over strong LLM-based baselines. We are also grateful that the reviewer finds the claims well supported and the work relevant to the TMLR audience. Below, we address the identified limitations and requested clarifications in detail.
>
> W1: Manual operator libraries & in-context examples limit scalability
>
> We acknowledge that TS-Reasoner uses a manually defined operator toolbox and a small set of annotated in-context examples; however, these components define only a minimal execution interface, not task-specific solutions. The operator library is deliberately compact and reusable, wrapping existing time series tools rather than encoding domain logic. Crucially, task-specific reasoning is not pre-engineered: it is generated at runtime through a single LLM-based custom operator that translates natural-language requirements (e.g., constraints or qualitative knowledge) into executable refinement or update code. As demonstrated in the generalization study (Table 4), this mechanism enables adaptation to new constraint types with little or no additional prompting. The in-context examples therefore act as lightweight bootstrapping for learning the operator interface, rather than per-task customization, and can be progressively reduced or replaced by automated refinement mechanisms in future work.
>
> W2: No direct modeling of raw time series data
>
> TS-Reasoner is not designed to replace time series foundation models, but to effectively leverage them. Rather than directly encoding raw temporal signals which we argue is a known limitation of LLMs (as demonstrated in all experiment results), the system delegates representation learning to specialized forecasting and analysis models (including foundation models such as Chronos and TimeGPT) and focuses on task decomposition, tool selection, and execution control. This separation enables model-agnostic integration and avoids coupling the framework to any specific backbone. In contrast, approaches that directly model time series within a multimodal backbone (e.g., ChatTime) offer limited interpretability due to opaque internal representations and, in our experiments, do not surpass TS-Reasoner in task success rate or inference quality.  The updated table 2 provides comparison with Chattime. Overall, TS-Reasoner complements rather than constrains progress in time series foundation models by providing a transparent and extensible reasoning layer on top of them.
>
> W3: Stability and interpretability of the task decomposer
>
> TS-Reasoner actually offers practical interpretability through its explicit operator-level programs. Each solution consists of a transparent sequence of steps shown in figure 3 (e.g., data retrieval, augmentation, forecasting, refinement), making the reasoning process significantly more interpretable than end-to-end multimodal models. The current design already improves interpretability relative to black-box alternatives.
> Stability is empirically evaluated via paraphrased-instruction experiments (TS-Reasoner w/ PQ) across all tasks (Table 2, Fig. 5). As further evidenced by the updated error analysis, TS-Reasoner maintains a stable success rate under instruction variation, with failures dominated by constraint violations (6.2–7.5%) rather than execution errors. The slightly higher violation rate under paraphrasing reflects increased ambiguity in constraint expression, rather than instability in task decomposition or execution.
>
> W4: Dataset scale
>
> We agree that the current benchmark does not fully capture all sources of noise and complexity encountered in industrial deployments. Our goal is not to provide a large pre-training or post-training corpus, but rather a procedurally complex reasoning benchmark that isolates and evaluates multi-step inference behavior. To complement this setting and assess generalization beyond the proposed tasks, we additionally include evaluation on established multimodal time series benchmarks used in prior work. Specifically, we report results on ChatTime’s benchmarks[2], including PTF context-guided forecasting and TSQA, in Appendix A. This additional evaluation complements our benchmark by expanding scale while preserving a focus on reasoning-oriented assessment.
>
> [2] Wang, Chengsen, et al. "Chattime: A unified multimodal time series foundation model bridging numerical and textual data." Proceedings of the AAAI Conference on Artificial Intelligence. Vol. 39. No. 12. 2025.

---

> ### Author Response · Authors · 2026-01-20
>
> We also summarize the newly added content to table 2 and figure 5 here.
> ## Table 2 updated baselines:
>
> ### Prediction w/ Covariates
> | Constraint      | Metric       | **ARIMA**     | **Chronos**   | **ChatTime**  | **TS-Reasoner**       |
> | --------------- | ------------ | ------------- | ------------- | ------------- | ------------- |
> | **Max Load**    | SR           | 0.7           | 0.8947        | 0.8421        | 1             |
> |                 | MAPE (Std) ↓ | 0.2237 (0.31) | 0.1502 (0.21) | 0.1275 (0.21) | 0.0621 (0.10) |
> | **Min Load**    | SR           | 0.8           | 0.85          | 0.9           | 1             |
> |                 | MAPE (Std) ↓ | 0.1579 (0.22) | 0.1205 (0.22) | 0.1158 (0.14) | 0.0564 (0.06) |
> | **Ramp Rate**   | SR           | 0.85          | 0.4444        | 0.6667        | 1             |
> |                 | MAPE (Std) ↓ | 0.1945 (0.25) | 0.2153 (0.31) | 0.2522 (0.34) | 0.0719 (0.20) |
> | **Variability** | SR           | 0.85          | 0.7778        | 0.7222        | 0.9444        |
> |                 | MAPE (Std) ↓ | 0.2805 (0.15) | 0.1223 (0.15) | 0.1725 (0.18) | 0.0577 (0.06) |
> | **Average**     | SR           | 0.8000        | 0.7417        | 0.7828        | **0.9861**    |
> |                 | MAPE ↓       | 0.2142        | 0.1521        | 0.1670        | *0.0620*      |
>
>
>
> ---
>
> ### Prediction w/o Covariates
>
> | Constraint      | Metric       | **ARIMA**     | **Chronos**   | **ChatTime**  | **TS-Reasoner**       |
> | --------------- | ------------ | ------------- | ------------- | ------------- | ------------- |
> | **Max Load**    | SR           | 1             | 0.9           | 0.95          | 1             |
> |                 | MAPE (Std) ↓ | 0.1457 (0.14) | 0.1579 (0.15) | 0.1207 (0.17) | 0.0799 (0.11) |
> | **Min Load**    | SR           | 0.95          | 0.9           | 0.75          | 1             |
> |                 | MAPE (Std) ↓ | 0.1741 (0.21) | 0.2142 (0.24) | 0.1151 (0.17) | 0.1366 (0.20) |
> | **Ramp Rate**   | SR           | 0.9           | 0.8           | 0.7           | 0.85          |
> |                 | MAPE (Std) ↓ | 0.1616 (0.11) | 0.2169 (0.16) | 0.1142 (0.07) | 0.1191 (0.17) |
> | **Variability** | SR           | 1             | 0.8           | 0.8           | 0.85          |
> |                 | MAPE (Std) ↓ | 0.1269 (0.04) | 0.1543 (0.05) | 0.0929 (0.04) | 0.0767 (0.04) |
> | **Average**     | SR           | 0.9625        | 0.8500        | 0.8000        | **0.9250**    |
> |                 | MAPE ↓       | 0.1521        | 0.1858        | 0.1107        | *0.1031*      |
>
>
>
> ---
>
> ### Prediction across Multiple Grids
>
> | Constraint      | Metric       | **ARIMA**     | **Chronos**   | **ChatTime**  | **TS-Reasoner**       |
> | --------------- | ------------ | ------------- | ------------- | ------------- | ------------- |
> | **Max Load**    | SR           | 0.55          | 0.85          | 0.8           | 0.9           |
> |                 | MAPE (Std) ↓ | 0.3975 (0.34) | 0.1324 (0.19) | 0.1309 (0.13) | 0.1491 (0.18) |
> | **Min Load**    | SR           | 0.6           | 0.8           | 0.85          | 1             |
> |                 | MAPE (Std) ↓ | 0.4087 (0.34) | 0.1031 (0.12) | 0.1411 (0.14) | 0.1614 (0.18) |
> | **Ramp Rate**   | SR           | 0.75          | 0.65          | 0.6           | 0.9           |
> |                 | MAPE (Std) ↓ | 0.5100 (0.31) | 0.1280 (0.07) | 0.1687 (0.08) | 0.1285 (0.13) |
> | **Variability** | SR           | 0.85          | 0.65          | 0.5           | 1             |
> |                 | MAPE (Std) ↓ | 0.3599 (0.28) | 0.1930 (0.22) | 0.2039 (0.25) | 0.1752 (0.25) |
> | **Average**     | SR           | 0.6875        | 0.7375        | 0.6875        | **0.9500**    |
> |                 | MAPE ↓       | 0.4190        | 0.1391        | 0.1612        | *0.1535*      |
>
>
>
> ---
>
> ## Figure 5 updated baselines:
>
> | Model                 | **Extreme Weather Detection w/ Reference Data** |        | **Extreme Weather Detection w/ Anomaly Rate** |            | **Causal Discovery** |            |
> | --------------------- | ------------------- | ------ | --------------------------- | ---------- | -------------------- | ---------- |
> |                       | SR ↑                | F1 ↑   | SR ↑                        | F1 ↑       | SR ↑                 | Accuracy ↑ |
> | **TS-Reasoner**       | **0.78**                | 0.9214 | **1.00**                    | **0.7569** | **1.00**             | **0.7915** |
> | **MATMCD**            | –                   | –      | –                           | –          | 0.50                 | 0.6294     |
> | **Z-Score**           | 0.08                | **0.9558** | 0.92                        | 0.1895     | –                    | –          |
> | **Granger Causality** | –                   | –      | –                           | –          | 0.78                 | 0.6460     |

---

### Review · Reviewer_8oWC · 2025-12-30

**Summary Of Contributions:**

In the paper the authors propose a multi-step paradigm to design a large language model backboned agent for generic time series tasks including forecasting, inference, anomality detection, etc. They create a new benchmark involving time series tasks requiring multi-step inference including constrained forecasting, anomaly detection and causal discovery. They benchmark their proposed agent entitled TS-Responsor along with language model baselines to demonstrate the significance of the introduction of multi-round coding based solutions.

**Audience:**

Yes

**Audience Explanation:**

The research question regarding general purposed time series agents studied in this submission is well-defined and critical. The method proposed in the submission follows common and empirically tested practices for developing LLM based agents. Researchers and pracitioners in related domains are expected to be interested.

**Claims And Evidence:**

No

**Claims Explanation:**

I'd like to split the submission into the proposed methodology, a new benchmark dataset and related benchmarking. The proposed methodology itself, which involves the construction of a LLM based agent mainly focusing on coding tools, follows standard sophisticated agent building strategies hence is relatively convincing. My major concerns are about the benchmark dataset and the benchmarking of the methodology.

The benchmark dataset is of a pretty narrow selections of domains, and the setup seems to be quite tailored to the proposed method.  The contribution of such a dataset to the community of multi-modal time series solutions is unclear given the current amount of baselines and benchmarking involved.

Regard the empirical benchmarking itself, I can see potentially improvements of
(1) including results on existing multi-modal benchmarks,
(2) introducing more relevant baseline methods
(3) including more ablation studies regarding other aspects of the proposed TS-Responsor

**Requested Changes:**

The requested changes are mostly ablation studies necessary to demonstrate the significance of the submission:

1. Benchmarking TS-Responsor with the relevant tasks from existing benchmarks (see, e.g., [1][2][3]) to establish more comparison w.r.t. existing methods.

2. More baseline methods should be included, such as
- a small selection of relevant statistical methods on all tasks.
- time series specific foundation models on forecasting tasks (e.g., Chronos).
- a few other milti-modal time series solution when applicable (see, e.g., this survey [4]).
- LLMs with paraphrasing if applicable.

3. If I understand correctly the selected operators are critical for TS-Responsor. It would be better to have much more ablation on this particular point, for example
- how and how often is each operator used.
- a detailed ablation on the inclusion / exclusion of key set or sets of operators.

 4. Consider reporting the cost of LLM calls.

[1] Williams, Andrew Robert, et al. "Context is Key: A Benchmark for Forecasting with Essential Textual Information." CoRR (2024).
[2] Wang, Chengsen, et al. "Chattime: A unified multimodal time series foundation model bridging numerical and textual data." Proceedings of the AAAI Conference on Artificial Intelligence. Vol. 39. No. 12. 2025.
[3] Liu, Haoxin, et al. "Time-mmd: Multi-domain multimodal dataset for time series analysis." Advances in Neural Information Processing Systems 37 (2024): 77888-77933.
[4] Jiang, Yushan, et al. "Multi-modal time series analysis: A tutorial and survey." Proceedings of the 31st ACM SIGKDD Conference on Knowledge Discovery and Data Mining V. 2. 2025.

---

> ### Author Response · Authors · 2026-01-19
>
> We thank the reviewer for the careful evaluation and constructive feedback. We appreciate the reviewer’s recognition that the research question of general-purpose time series agents is well defined, and that the proposed TS-Reasoner methodology follows empirically grounded and effective agent-design practices. We are also encouraged that the reviewer finds the overall paradigm convincing and relevant to both researchers and practitioners. Below, we address the reviewer’s concerns regarding the benchmark design, baselines, and ablation studies in detail.
>
> 1) Dataset contribution
>
> We acknowledge that the benchmark currently spans a limited set of application domains; however, its contribution is not intended to provide broad domain coverage, but to introduce and operationalize a previously under-specified task setting: multi-step time series inference with explicit evaluation of validity and quality. The benchmark is defined at the level of task structure and evaluation criteria (e.g., constraint satisfaction), and does not assume any particular modeling paradigm, tool abstraction, or agent design. While TS-Reasoner provides one concrete solution, the tasks can also be addressed by alternative approaches, such as direct code-based agents or multimodal time series models. Importantly, the dataset is generated through a dynamic synthesis pipeline, enabling systematic scaling in size and extension to additional domains with minimal annotation of natural-language task instructions and specifications (e.g., constraints or domain knowledge).
> Most existing multimodal time series benchmarks primarily focus on context-aided forecasting, where auxiliary information influences prediction accuracy. In contrast, our benchmark includes diverse task formulations, such as diagnostic analysis and causal reasoning, and further evaluates constraint-aware forecasting scenarios that commonly arise in real-world practice but are largely absent from existing multimodal benchmarks. Rather than serving as a pre-training or post-training corpus, the benchmark is intended as a downstream evaluation suite for multimodal time series models. We have revised Section 4 of the manuscript to more clearly articulate this task-centric motivation and its relevance to the broader multimodal and agent-based time series community.
>
> 2) “Need results on existing multimodal benchmarks + more relevant baselines”
>
> We appreciate this request and have strengthened the empirical benchmarking accordingly. Beyond our proposed multi-step tasks, we include additional evaluation on established benchmarks used in prior multimodal time-series work, reporting results on ChatTime’s benchmarks[2] (PTF context-guided forecasting and TSQA) in the appendix section A. We also enriched the suggested baseline coverage: (i) classical statistical methods (ARIMA, Granger causality, Z-score detection) are included as task-appropriate references, (ii) time series foundation models such as Chronos are used for forecasting baselines where applicable, and (iii) multimodal baselines (e.g., ChatTime[2], MATMCD[4][5]) are included when the task setting matches their input modality. We have updated our experiment results section(table 2 and figure 5) with more baseline comparisons and the appendix section A with additional benchmark results (table 5 and table 6)
> [5]Shen, ChengAo, et al. "Exploring multi-modal integration with tool-augmented llm agents for precise causal discovery." arXiv preprint arXiv:2412.13667 1.3 (2024).

---

> ### Author Response · Authors · 2026-01-19
>
> We also summarize the newly added content to table 2 and figure 5 here.
> ## Table 2 updated baselines:
>
> ### Prediction w/ Covariates
> | Constraint      | Metric       | **ARIMA**     | **Chronos**   | **ChatTime**  | **TS-Reasoner**       |
> | --------------- | ------------ | ------------- | ------------- | ------------- | ------------- |
> | **Max Load**    | SR           | 0.7           | 0.8947        | 0.8421        | 1             |
> |                 | MAPE (Std) ↓ | 0.2237 (0.31) | 0.1502 (0.21) | 0.1275 (0.21) | 0.0621 (0.10) |
> | **Min Load**    | SR           | 0.8           | 0.85          | 0.9           | 1             |
> |                 | MAPE (Std) ↓ | 0.1579 (0.22) | 0.1205 (0.22) | 0.1158 (0.14) | 0.0564 (0.06) |
> | **Ramp Rate**   | SR           | 0.85          | 0.4444        | 0.6667        | 1             |
> |                 | MAPE (Std) ↓ | 0.1945 (0.25) | 0.2153 (0.31) | 0.2522 (0.34) | 0.0719 (0.20) |
> | **Variability** | SR           | 0.85          | 0.7778        | 0.7222        | 0.9444        |
> |                 | MAPE (Std) ↓ | 0.2805 (0.15) | 0.1223 (0.15) | 0.1725 (0.18) | 0.0577 (0.06) |
> | **Average**     | SR           | 0.8000        | 0.7417        | 0.7828        | **0.9861**    |
> |                 | MAPE ↓       | 0.2142        | 0.1521        | 0.1670        | *0.0620*      |
>
>
>
> ---
>
> ### Prediction w/o Covariates
>
> | Constraint      | Metric       | **ARIMA**     | **Chronos**   | **ChatTime**  | **TS-Reasoner**       |
> | --------------- | ------------ | ------------- | ------------- | ------------- | ------------- |
> | **Max Load**    | SR           | 1             | 0.9           | 0.95          | 1             |
> |                 | MAPE (Std) ↓ | 0.1457 (0.14) | 0.1579 (0.15) | 0.1207 (0.17) | 0.0799 (0.11) |
> | **Min Load**    | SR           | 0.95          | 0.9           | 0.75          | 1             |
> |                 | MAPE (Std) ↓ | 0.1741 (0.21) | 0.2142 (0.24) | 0.1151 (0.17) | 0.1366 (0.20) |
> | **Ramp Rate**   | SR           | 0.9           | 0.8           | 0.7           | 0.85          |
> |                 | MAPE (Std) ↓ | 0.1616 (0.11) | 0.2169 (0.16) | 0.1142 (0.07) | 0.1191 (0.17) |
> | **Variability** | SR           | 1             | 0.8           | 0.8           | 0.85          |
> |                 | MAPE (Std) ↓ | 0.1269 (0.04) | 0.1543 (0.05) | 0.0929 (0.04) | 0.0767 (0.04) |
> | **Average**     | SR           | 0.9625        | 0.8500        | 0.8000        | **0.9250**    |
> |                 | MAPE ↓       | 0.1521        | 0.1858        | 0.1107        | *0.1031*      |
>
>
>
> ---
>
> ### Prediction across Multiple Grids
>
> | Constraint      | Metric       | **ARIMA**     | **Chronos**   | **ChatTime**  | **TS-Reasoner**       |
> | --------------- | ------------ | ------------- | ------------- | ------------- | ------------- |
> | **Max Load**    | SR           | 0.55          | 0.85          | 0.8           | 0.9           |
> |                 | MAPE (Std) ↓ | 0.3975 (0.34) | 0.1324 (0.19) | 0.1309 (0.13) | 0.1491 (0.18) |
> | **Min Load**    | SR           | 0.6           | 0.8           | 0.85          | 1             |
> |                 | MAPE (Std) ↓ | 0.4087 (0.34) | 0.1031 (0.12) | 0.1411 (0.14) | 0.1614 (0.18) |
> | **Ramp Rate**   | SR           | 0.75          | 0.65          | 0.6           | 0.9           |
> |                 | MAPE (Std) ↓ | 0.5100 (0.31) | 0.1280 (0.07) | 0.1687 (0.08) | 0.1285 (0.13) |
> | **Variability** | SR           | 0.85          | 0.65          | 0.5           | 1             |
> |                 | MAPE (Std) ↓ | 0.3599 (0.28) | 0.1930 (0.22) | 0.2039 (0.25) | 0.1752 (0.25) |
> | **Average**     | SR           | 0.6875        | 0.7375        | 0.6875        | **0.9500**    |
> |                 | MAPE ↓       | 0.4190        | 0.1391        | 0.1612        | *0.1535*      |
>
>
>
> ---
>
> ## Figure 5 updated baselines:
>
> | Model                 | **Extreme Weather Detection w/ Reference Data** |        | **Extreme Weather Detection w/ Anomaly Rate** |            | **Causal Discovery** |            |
> | --------------------- | ------------------- | ------ | --------------------------- | ---------- | -------------------- | ---------- |
> |                       | SR ↑                | F1 ↑   | SR ↑                        | F1 ↑       | SR ↑                 | Accuracy ↑ |
> | **TS-Reasoner**       | **0.78**                | 0.9214 | **1.00**                    | **0.7569** | **1.00**             | **0.7915** |
> | **MATMCD**            | –                   | –      | –                           | –          | 0.50                 | 0.6294     |
> | **Z-Score**           | 0.08                | **0.9558** | 0.92                        | 0.1895     | –                    | –          |
> | **Granger Causality** | –                   | –      | –                           | –          | 0.78                 | 0.6460     |

---

> ### Author Response · Authors · 2026-01-19
>
> ## Table 5: Time Series Question Answering (Accuracy ↑)
>
> > *Higher is better.*
>
> ### Feature-wise Accuracy Across Sequence Lengths
>
> | Feat         | Len | GPT4   | GPT3.5 | GLM4   | LLaMA3 | ChatTime | **TS-Reasoner** |
> | ------------ | --- | ------ | ------ | ------ | ------ | -------- | -------------- |
> | Trend        | 64  | 0.6532 | 0.3507 | 0.7319 | 0.6799 | 0.9011   | **0.9967**     |
> |              | 128 | 0.7015 | 0.5846 | 0.7574 | 0.5855 | 0.9068   | **0.9927**     |
> |              | 256 | 0.7482 | 0.5028 | 0.6377 | 0.6143 | 0.8843   | **0.9687**     |
> |              | 512 | 0.6346 | 0.5903 | 0.6697 | 0.6753 | 0.8234   | **0.9343**     |
> | Volatility   | 64  | 0.5585 | 0.5633 | 0.6797 | 0.6373 | 0.7874   | **0.8900**     |
> |              | 128 | 0.4979 | 0.3839 | 0.4770 | 0.4756 | 0.6954   | **0.9803**     |
> |              | 256 | 0.4624 | 0.4894 | 0.5418 | 0.5246 | 0.6228   | **0.9713**     |
> |              | 512 | 0.3169 | 0.3796 | 0.4549 | 0.5261 | 0.5736   | **0.6493**     |
> | Season       | 64  | 0.3518 | 0.3428 | 0.3366 | 0.3484 | 0.6639   | **0.8627**     |
> |              | 128 | 0.3515 | 0.3952 | 0.3464 | 0.3958 | 0.6517   | **0.8767**     |
> |              | 256 | 0.5283 | 0.5089 | 0.3892 | 0.4120 | 0.6463   | **0.8810**     |
> |              | 512 | 0.4457 | 0.4889 | 0.3892 | 0.4127 | 0.6244   | **0.8890**     |
> | Outlier      | 64  | 0.7230 | 0.4325 | 0.5359 | 0.7051 | 0.8773   | **0.8787**     |
> |              | 128 | 0.6327 | 0.5940 | 0.5298 | 0.5694 | 0.9032   | **0.9827**     |
> |              | 256 | 0.6795 | 0.4579 | 0.5019 | 0.5073 | 0.8593   | **0.9913**     |
> |              | 512 | 0.6219 | 0.4996 | 0.2822 | 0.4085 | 0.7478   | **0.9910**     |
> | **Avg. Acc** |     | 0.5567 | 0.4728 | 0.5163 | 0.5299 | 0.7605   | **0.9210**     |
>
> ---
>
> ## Table 6: PTF Context-Guided Forecasting (MAE ↓)
>
> > *Lower is better.*
>
> ### Dataset-Specific Forecast vs. Dataset-Shared Forecast
>
> | Hist         | Pred | DLinear | GPT4TS | TimeLLM | TGForecaster | Moirai | TimesFM | Chronos | ChatTime | **TS-Reasoner** |
> | ------------ | ---- | ------- | ------ | ------- | ------------ | ------ | ------- | ------- | -------- | -------------- |
> | 48           | 24   | 0.5204  | 0.4373 | 0.4211  | 0.4411       | 0.5981 | 0.4851  | 0.4813  | 0.4849   | **0.4194**     |
> | 72           | 24   | 0.5075  | 0.4253 | 0.4031  | 0.3943       | 0.5776 | 0.4258  | 0.4276  | 0.4307   | **0.3619**     |
> | 96           | 24   | 0.4965  | 0.3921 | 0.4392  | 0.3653       | 0.5179 | 0.4054  | 0.4336  | 0.3920   | **0.3538**     |
> | 120          | 24   | 0.4796  | 0.3713 | 0.3594  | 0.3594       | 0.5245 | 0.3807  | 0.3902  | 0.3480   | **0.3336**     |
> | **Avg. MAE** |      | 0.5010  | 0.4065 | 0.4057  | 0.3900       | 0.5545 | 0.4243  | 0.4332  | 0.4139   | **0.3672**     |
>
> ---
>
> Table 6 presents results on the TSQA benchmark, which evaluates question answering accuracy across feature groups (trend, volatility, seasonality, and outliers) and increasing sequence lengths (64/128/256/512). A notable difference between the two methods is their behavior as sequence length increases. ChatTime’s performance generally degrades with longer sequences across most feature groups, whereas TS-Reasoner maintains stable accuracy as sequence length grows with the exception of Volatility. This suggests that TS-Reasoner is less constrained by sequence length due to its operator-based execution and decomposition strategy, rather than relying on direct sequence encoding within the language model. We further evaluate TS-Reasoner on established benchmarks from the ChatTime framework to assess performance beyond our proposed multi-step inference tasks. Table 5 reports results on the PTF dataset for context-guided forecasting. TS-Reasoner achieves lower MAE than ChatTime across all evaluated history lengths.

---

> ### Author Response · Authors · 2026-01-20
>
> 3) “More ablations on operators: how often used; inclusion/exclusion of key sets”
>
> We agree that operator contribution is central to TS-Reasoner. In addition to the component ablations, we include a targeted operator ablation (Table 3) comparing the removal of the custom refinement operator and external data retrieval operators, which reveals distinct effects on success rate versus inference quality. Specifically, custom operators are critical for ensuring successful task completion under constraints, whereas retrieval operators primarily contribute to enriching contextual information and improving prediction accuracy.
>
> To further analyze operator roles within forecasting tasks, we report operator invocation statistics. For forecasting tasks, the LLM-generated refinement operator is invoked in nearly all runs (97%), underscoring its importance for enforcing task-specific constraints and revising predictions. External data retrieval operators are used in 32% of forecasting runs, primarily in cases where the task description indicates a lack of contextual information that could be beneficial; in these scenarios, the model identifies the need for additional context and retrieves relevant covariates to enrich the input before forecasting. Regarding prediction models, multivariate forecasting with covariates (MultiPreOP) is selected in approximately 66% of forecasting workflows, while univariate forecasting is used otherwise. Overall, execution traces reveal a common multi-step pattern (often involving retrieval, augmentation, forecasting, and refinement), while allowing flexibility depending on task requirements.
>
> ## Table 3: targeted operator inclusion/exclusion ablation study.
> | Constraint Types     | **TS-Reasoner** |            | **No Custom Operator** |        | **No Retrieval Operator** |        |
> | -------------------- | -------------- | ---------- | ---------------------- | ------ | ------------------------- | ------ |
> |                      | SR ↑           | MAPE ↓     | SR ↑                   | MAPE ↓ | SR ↑                      | MAPE ↓ |
> | **Max Load**         | 1.00           | 0.0799     | 0.85                   | 0.0557 | 1.00                      | 0.1349 |
> | **Min Load**         | 1.00           | 0.1366     | 0.80                   | 0.1559 | 1.00                      | 0.1723 |
> | **Load Ramp Rate**   | 0.85           | 0.1191     | 0.15                   | 0.1193 | 0.95                      | 0.1707 |
> | **Load Variability** | 0.85           | 0.0767     | 0.80                   | 0.0843 | 1.00                      | 0.1233 |
> | **Average**          | **0.925**      | **0.1031** | 0.65                   | 0.1038 | 0.9875                    | 0.1503 |
>
>
>
> 4) "Consider reporting the cost of LLM calls"
>
> We estimate inference cost by explicitly separating new input tokens, cached input tokens, and output tokens in the multi-turn interaction. On average, each interaction turn produces 63 output tokens, priced at \\$10 per million tokens. With a maximum of 5 turns, the output cost is approximately
>
> 63 × 5 × 10 / 1,000,000 ≈ \\$0.003.
>
> The initial input prompt contains 2,647 tokens on average and is billed at the new-token rate of \\$2.5 per million tokens. In subsequent turns, previously seen inputs are reused and charged at the cached-token rate of \\$1.25 per million tokens. Accounting for token reuse across turns, the cumulative cached-input cost is approximately
>
> 2647 × (5 + 4 + 3 + 2) × 1.25 / 1,000,000 ≈ \\$0.49.
>
> Overall, the total inference cost is approximately \\$0.5 per question under this multi-turn setting. Importantly, reliance on proprietary models is not essential. We explicitly evaluate TS-Reasoner using open-source backbones (e.g., LLaMA-3.1-70B) and observe comparable performance, with task success rates of 94.83% vs. 94.25% and similar inference quality, as reported in Table 7 (Appendix).

---

### Review · Reviewer_qH5M · 2026-01-05

**Summary Of Contributions:**

This paper introduces TS-Reasoner, an agentic workflow for time series, with a corresponding toolkit. TS-Reasoner comprises an LLM that decomposes a task into subtasks, which are then implemented in a type of domain-specific language in the form of a toolkit. The LLM learns this DSL through in-context learning. If the produced code fails in some way, the plan is re-generated with the feedback produced by the error. This is extended to "multi-step" time series inference tasks, for which a novel dataset is constructed.

In general, the paradigm of LLMs for automated analysis is timely, and the general idea of building toolkits for agents makes sense. However, I have several concerns with the manuscript.

**Novelty & Framing** Please clarify the novelty of TS-Reasoner -- using ICL for tool-calling, with some feedback loop to fix errors, seems somewhat standard to me. It seems more reasonable to me that the work is centered on providing a toolkit that is simple and effective -- and as a bonus, quite compatible with LLM tooling -- but this is not how the manuscript is currently framed.

Notably, there are several existing and uncited workflows that incorporate domain knowledge, contextualization, and LLMs in time series prediction. For example, [1] summarizes and incorporates news in forecasting objectives. [2] provides additional methods for contextualizing and predicting with LLMs. [3] provides a fairly extensive survey on the "Agent-As-Data-Analyst" paradigm, including a section on time series and an NL2Code paradigm.

**Regarding Causality** The tool `CausalMatrixOP` refers to Granger causality explicitly, but Granger causality does not otherwise appear in the manuscript. Causal discovery for time series is quite a bit broader than just Granger causality, with substantively different methods, assumptions, and answers (c.f., [4]). Please clarify what paradigm is being used.

**Regarding Program Complexity** A premise of the paper is that TS-Reasoner is performing some type of multi-step reasoning, but example generated programs are quite simple. For example, the "causal reasoning" example in Figure 3 essentially just calls the wrapper `CausalMatrixOP`, with the extreme weather identification example similarly just calling `AnomalDetOP`. From these examples, it feels that writing the DSL code directly could even be easier/faster than writing a prompt + waiting for LLM generation.

**Feedback Mechanism** The text surrounding the feedback mechanism is quite vague, especially regarding model selection. It's stated that "conditioned on such support from the evaluation benchmark, TS-Reasoner leverages feedback on prediction quality (e.g., MAPE) to experiment with different model choices and select the best-performing one." This raises important methodological questions, e.g., on what data is prediction quality evaluated on? If it's on training data, this can suggest overfitting; if it's on test data, this suggests data leakage; if it's on validation data, how is this validation data specified?

**Baselines** The baselines are a bit unclear to me in Section 5.2. In particular, what libraries do the baseline models have access to? How much prompt engineering was done?

**Writing** I find the writing generally clear, but there are many typos and editorial issues (see Requested Changes below). The manuscript requires a careful read to fix these.

**Additional Comments:**

# References

[1] Wang, X., Feng, M., Qiu, J., Gu, J., & Zhao, J. (2024). From news to forecast: Integrating event analysis in LLM-based time series forecasting with reflection. _Advances in Neural Information Processing Systems_, _37_, 58118-58153.

[2] Lee, G., Yu, W., Shin, K., Cheng, W., & Chen, H. (2025, April). Timecap: Learning to contextualize, augment, and predict time series events with large language model agents. In _Proceedings of the AAAI Conference on Artificial Intelligence_ (Vol. 39, No. 17, pp. 18082-18090).

[3] Tang, Z., Wang, W., Zhou, Z., Jiao, Y., Xu, B., Niu, B., ... & Wu, F. (2025). LLM/Agent-As-Data-Analyst: A Survey. _arXiv preprint arXiv:2509.23988_.

[4] Assaad, C. K., Devijver, E., & Gaussier, E. (2022). Survey and evaluation of causal discovery methods for time series. _Journal of Artificial Intelligence Research_, _73_, 767-819.

**Audience:**

No

**Audience Explanation:**

I'm skeptical that this submission is of interest to a subset of the TMLR audience, as-written. In my opinion, there is a potentially interesting library developed here, but its connection to an LLM doesn't seem to constitute a novel agentic framework. Also of note for this question are the many typos, and lack of open-source access to the toolkit (please do point me to this if I missed it in the original submission).

**Broader Impact Concerns:**

I believe a broader impact statement regarding the limitations of the current models and benchmarking would be helpful to some subset of potential readers, in particular, applied ML practitioners.

**Claims And Evidence:**

No

**Claims Explanation:**

In addition to the above concerns (regarding feedback mechanism specification, and baselines):
- It's stated that "TS-Reasoner dominates all baseline models in this aspect, highlighting its superior ability to integrate domain knowledge, and perform structured reasoning over complex time series analysis tasks." There are two potential issues here:
	- In extreme weather detection w/ reference data, TSR is not the best with respect to F1 score.
	- I'm not convinced this illustrates "a superior ability to integrate domain knowledge," rather than access to a superior time series library. Indeed, this seems to be suggested in the ablation study.
- Are custom operators ever used in results? If so, how? `RefGenOP` appears in Figure 3, but uses an undefined `refine_requirement`.
	- In a footnote, it is written that "We also share a few sample questions here https://anonymous.4open.science/r/sample_dataset/." This link contains only a pkl file; this is, in general, an unsafe file format to load from an unknown source. Therefore, I did not check it. Please, upload these in a safer file format.

**Requested Changes:**

Critical:
- Clarify novelty, as discussed above. In particular, please discuss related work such as [1, 2] (and potentially others from [3]) in more depth.
- Clarify how causality is to be interpreted, as above.
- Clarify the feedback mechanism for model selection, as above.
- Amend or provide additional evidence for statements about "integrating domain knowledge," as discussed in the Evidence section.

Minor:
- Pie charts are used extensively, but are quite poor for data visualization purposes. Please consider using an alternative type of chart, e.g., stacked bar charts.
- It's stated that "the upper-right corner [of Figure 5] represents the ideal problem solver that achieves both a high success rate and high inference quality among successful cases." But this isn't true because of axis limits. Please revise the statement.

I additionally have many editorial remarks that should be fixed:
- Please use `\citep` where appropriate. Currently, many references appear as in-text citations when they should actually be parenthetical (e.g., in Section 1).
- "Using expert demonstration of tool usage in-context learning (Brown et al., 2020)" in Section 2 reads awkwardly, please revise.
- In Section 2, the heading "Task Decomposer" should not be used as part of the sentence.
- In Section 2, "volatility, or evaluating distribution properties.3) External Data Retrieval Operators" is missing a space between "properties." and "3)".
- In the "Question Template (Causal Discovery)" box, there are missing brackets around tjhe second `{variable names}`.
- In Section 4.2, elements of the list of electricity grids should have spaces between them.
- On page 8, "TS-Reasoner 's" should not have a space.
- Section 5.2, "novel new paradigm" is redundant. Use "novel" or "new."
- Section 5.2, "Python" should be capitalized.
- Section 5.2, "Among successful task completions,TS-Reasoner demonstrates the" is missing a space.
- Section 5.2, "causal discovry" should be "causal discovery".
- Appendix C, "Granger" should be capitalized in "granger causality."

As there are many editorial remarks, I suspect this list is not exhaustive, and the manuscript should be carefully checked.

---

> ### Author Response · Authors · 2026-01-19
>
> We thank the reviewer for their careful reading and detailed feedback. We appreciate the reviewer’s recognition that the problem addressed by TS-Reasoner is timely and that the paradigm of toolkit-based, LLM-assisted analysis for time series is well motivated and practically meaningful. We also value the reviewer’s constructive comments on framing, methodology, and evaluation, which help clarify the contributions of this work. Below, we address each concern in detail.
>
> 1. Novelty framing
>
> We appreciate the reviewer’s comments and agree that generic agent mechanisms such as in-context tool calling and execution feedback are not, by themselves, novel. Our intention was not to claim novelty in these mechanisms, but rather to introduce a domain-specialized system and task formulation for time series analysis, which we have now clarified throughout the manuscript.
> In the revised paper, we explicitly frame our contribution around multi-step time series inference as a task paradigm, motivated by the observation that real-world time series analysis often requires procedural reasoning, constraint handling, and dynamic workflow composition. There are capabilities not captured by existing fixed input–output benchmarks or foundation models. Correspondingly, TS-Reasoner is presented as a domain-oriented instantiation of a tool-augmented LLM, where the primary contribution lies in (i) a standardized operator abstraction for time series analysis, (ii) structured programmatic execution to ensure numerical reliability, and (iii) a benchmark and evaluation protocol designed to measure multi-step inference rather than single-step prediction accuracy. Standard execution feedback is used as an enabling mechanism for robustness, not as a core novelty claim.
> We have also strengthened the Related Work section to better situate our contribution. In particular, we now explicitly discuss recent LLM-based time series agent systems that incorporate external context or event information for forecasting and event prediction (e.g., Wang et al. [1]; Lee et al. [2]). We clarify that these works primarily target specific predictive tasks or model architectures, whereas our focus is on general multi-step time series inference workflows that span predictive and diagnostic tasks and require constraint-aware operator composition. More broadly, we position TS-Reasoner as a time-series–specific instantiation of the Agent-as-Data-Analyst paradigm surveyed by Tang et al. [3], with dedicated operators and evaluation protocols tailored to time series reasoning.
> These clarifications are reflected in revisions to the Introduction, Related Work, and contribution statements, ensuring that the paper’s novelty is clearly grounded in task formulation, domain-oriented tooling, and evaluation, rather than in generic agent scaffolding.
>
> 2. Causal paradigm
>
> Our causal tasks are synthetically generated, with ground-truth labels defined using a Granger-style predictive notion of temporal influence. This choice is made for practicality: Granger causality provides an observational, temporally grounded, and automatically verifiable criterion that is well suited for synthetic task generation and workflow-level evaluation. While Granger causality is increasingly viewed as capturing predictive influence rather than true causation, this aligns with our objective, as the goal of the causal task is not counterfactual reasoning but to assess whether an agent can correctly invoke, interpret, and integrate a causal analysis operator within a multi-step inference workflow. In many time series applications, such predictive causal notions are commonly used for forecasting, feature selection, and model design, where identifying variables that improve prediction is the primary concern.
> Alternative causal paradigms for time series such as structured causal models (SCMs), constraint-based methods based on conditional independence assumptions, score-based graph optimization approaches, or recent gradient-based formulations, rely on stronger assumptions, interventional data, or substantially higher computational complexity, making them difficult to control and evaluate consistently in the current synthetic setting[4]. We do not aim to benchmark general causal discovery algorithms under these paradigms; instead, our focus is on evaluating agentic reasoning and tool orchestration. Extending the benchmark to support richer causal formulations, including SCM-based or constraint-based causal reasoning, is a meaningful direction for future work. We have updated the causal mining task definition in section 4.1 to clarify this scope and motivation.

---

> ### Author Response · Authors · 2026-01-19
>
> 3. Feedback mechanism
>
> TS-Reasoner uses within-instance, temporally valid self-evaluation for intermediate feedback. For each input time series, the last 10% of observations is truncated as a pseudo-holdout, and prediction error on this segment (e.g., MAPE) is used only to guide workflow refinement or operator selection. The final evaluation is performed exclusively on the true future horizon, which is never exposed to feedback. Since feedback is temporally preceding, does not involve parameter updates, and does not persist across samples, this mechanism avoids both overfitting and data leakage. We have modified the feedback mechanism subsection in section 3 to clarify this.
>
> 4. Amend over-claiming statement
>
> We agree the original sentence should be more precise. In Extreme Weather Detection w/ Reference Data, while some baselines achieve a higher F1, they do so with substantially lower success rates, and F1 is computed only over successful cases. Consequently, a higher F1 for a baseline does not indicate better overall performance when end-to-end task completion is taken into account. We will revise the text to explicitly state that TS-Reasoner’s advantage is in achieving the best overall tradeoff (high success rate with competitive quality among successes), rather than claiming it is strictly best on F1. We have also removed the incorrect implication that the plot top right corner itself is the ideal solver but explicitly mention methods positioned toward the upper-right direction are preferred.
> We also agree that the performance gains should not be attributed solely to “domain knowledge integration.” Our ablation results indicate that improvements are driven primarily by structured workflow execution and domain-specific operators (i.e., tool specialization), with feedback mechanisms further improving reliability. We have revised the wording accordingly in section 5.2’s diagnostic experiments to emphasize tool-based structured reasoning and robust execution, and treat domain knowledge as one contributing component rather than the sole explanation.
> We thank the reviewer for pointing this out and agree that the presentation in Figure 3 can be clearer. Custom operators are used in the reported experiments, primarily for refinement. In particular, in constraint-aware electricity forecasting tasks, TS-Reasoner invokes RefGenOP 97\% of the time to generate a task-specific refinement function that adjusts predictions according to constraint specified in the question. In Figure 3, refine_requirement was intended to denote a placeholder for a natural-language refinement specification derived from either the original question or feedback; to avoid ambiguity, we have revised figure 3 to explicitly write {refine_requirement} and clarify that it represents a textual input passed to RefGenOP, rather than a predefined variable or hidden function.
>
> 5. Program complexity
>
> The observation that some example programs are compact is valid; however, this does not imply that the workflows can be hard-coded in advance. The DSL used by TS-Reasoner is intentionally underspecified, and a decent amount of effort is required to instantiate it correctly from natural-language instructions. For example, in the constrained forecasting task without covariates task (results reported in table2), the task decomposer learns to infer which covariates are relevant to the task, issue retrieval calls accordingly, and, if a requested variable is unavailable, interpret error feedback to revise its choice of covariates. Similarly, in causal analysis with domain knowledge task (results reported in table 4), the system must interpret qualitative causal statements (e.g., “advertising spend positively affects revenue”) and generate custom refinement code to update the causal graph accordingly. In constraint-aware electricity forecasting, the decomposer must formulate a task-specific refinement prompt that encodes operational constraints, which is then used by a custom operator to generate executable refinement logic. Thus, the complexity of TS-Reasoner lies not in producing long programs, but in deciding how to instantiate and refine operator-level workflows, which cannot be achieved by hard-coding DSL templates.

---

> ### Author Response · Authors · 2026-01-19
>
> 6. Clarification
>
> Baseline LLM agents are provided with the same task instructions and inputs as TS-Reasoner. For CodeAct agents, we modify the prompt to explicitly specify which variables are available and what information they contain, and propagate execution errors back to the agent following the standard CodeAct setup. For ReAct agents, we follow their multi-turn interaction protocol and include a reflection step prior to reasoning and code generation. All baseline agents have access to standard machine learning and data processing libraries (e.g., statsmodels, scikit-learn, Granger causality utilities). In addition, we include more task-specific baselines such as multimodal time series models Chattime, time series foundation models Chronos, and simple statistical methods ARIMA for each task to further enrich the comparison. Please see updated table 2 and figure 5.
>
> ## Table 2 updated baselines:
>
> ### Prediction w/ Covariates
> | Constraint      | Metric       | **ARIMA**     | **Chronos**   | **ChatTime**  | **TS-Reasoner**       |
> | --------------- | ------------ | ------------- | ------------- | ------------- | ------------- |
> | **Max Load**    | SR           | 0.7           | 0.8947        | 0.8421        | 1             |
> |                 | MAPE (Std) ↓ | 0.2237 (0.31) | 0.1502 (0.21) | 0.1275 (0.21) | 0.0621 (0.10) |
> | **Min Load**    | SR           | 0.8           | 0.85          | 0.9           | 1             |
> |                 | MAPE (Std) ↓ | 0.1579 (0.22) | 0.1205 (0.22) | 0.1158 (0.14) | 0.0564 (0.06) |
> | **Ramp Rate**   | SR           | 0.85          | 0.4444        | 0.6667        | 1             |
> |                 | MAPE (Std) ↓ | 0.1945 (0.25) | 0.2153 (0.31) | 0.2522 (0.34) | 0.0719 (0.20) |
> | **Variability** | SR           | 0.85          | 0.7778        | 0.7222        | 0.9444        |
> |                 | MAPE (Std) ↓ | 0.2805 (0.15) | 0.1223 (0.15) | 0.1725 (0.18) | 0.0577 (0.06) |
> | **Average**     | SR           | 0.8000        | 0.7417        | 0.7828        | **0.9861**    |
> |                 | MAPE ↓       | 0.2142        | 0.1521        | 0.1670        | *0.0620*      |
>
>
>
> ---
>
> ### Prediction w/o Covariates
>
> | Constraint      | Metric       | **ARIMA**     | **Chronos**   | **ChatTime**  | **TS-Reasoner**       |
> | --------------- | ------------ | ------------- | ------------- | ------------- | ------------- |
> | **Max Load**    | SR           | 1             | 0.9           | 0.95          | 1             |
> |                 | MAPE (Std) ↓ | 0.1457 (0.14) | 0.1579 (0.15) | 0.1207 (0.17) | 0.0799 (0.11) |
> | **Min Load**    | SR           | 0.95          | 0.9           | 0.75          | 1             |
> |                 | MAPE (Std) ↓ | 0.1741 (0.21) | 0.2142 (0.24) | 0.1151 (0.17) | 0.1366 (0.20) |
> | **Ramp Rate**   | SR           | 0.9           | 0.8           | 0.7           | 0.85          |
> |                 | MAPE (Std) ↓ | 0.1616 (0.11) | 0.2169 (0.16) | 0.1142 (0.07) | 0.1191 (0.17) |
> | **Variability** | SR           | 1             | 0.8           | 0.8           | 0.85          |
> |                 | MAPE (Std) ↓ | 0.1269 (0.04) | 0.1543 (0.05) | 0.0929 (0.04) | 0.0767 (0.04) |
> | **Average**     | SR           | 0.9625        | 0.8500        | 0.8000        | **0.9250**    |
> |                 | MAPE ↓       | 0.1521        | 0.1858        | 0.1107        | *0.1031*      |
>
>
>
> ---
>
> ### Prediction across Multiple Grids
>
> | Constraint      | Metric       | **ARIMA**     | **Chronos**   | **ChatTime**  | **TS-Reasoner**       |
> | --------------- | ------------ | ------------- | ------------- | ------------- | ------------- |
> | **Max Load**    | SR           | 0.55          | 0.85          | 0.8           | 0.9           |
> |                 | MAPE (Std) ↓ | 0.3975 (0.34) | 0.1324 (0.19) | 0.1309 (0.13) | 0.1491 (0.18) |
> | **Min Load**    | SR           | 0.6           | 0.8           | 0.85          | 1             |
> |                 | MAPE (Std) ↓ | 0.4087 (0.34) | 0.1031 (0.12) | 0.1411 (0.14) | 0.1614 (0.18) |
> | **Ramp Rate**   | SR           | 0.75          | 0.65          | 0.6           | 0.9           |
> |                 | MAPE (Std) ↓ | 0.5100 (0.31) | 0.1280 (0.07) | 0.1687 (0.08) | 0.1285 (0.13) |
> | **Variability** | SR           | 0.85          | 0.65          | 0.5           | 1             |
> |                 | MAPE (Std) ↓ | 0.3599 (0.28) | 0.1930 (0.22) | 0.2039 (0.25) | 0.1752 (0.25) |
> | **Average**     | SR           | 0.6875        | 0.7375        | 0.6875        | **0.9500**    |
> |                 | MAPE ↓       | 0.4190        | 0.1391        | 0.1612        | *0.1535*      |
>
>
>
> ---

---

> ### Author Response · Authors · 2026-01-19
>
> ## Figure 5 updated baselines:
>
> | Model                 | **Extreme Weather Detection w/ Reference Data** |        | **Extreme Weather Detection w/ Anomaly Rate** |            | **Causal Discovery** |            |
> | --------------------- | ------------------- | ------ | --------------------------- | ---------- | -------------------- | ---------- |
> |                       | SR ↑                | F1 ↑   | SR ↑                        | F1 ↑       | SR ↑                 | Accuracy ↑ |
> | **TS-Reasoner**       | **0.78**                | 0.9214 | **1.00**                    | **0.7569** | **1.00**             | **0.7915** |
> | **MATMCD**            | –                   | –      | –                           | –          | 0.50                 | 0.6294     |
> | **Z-Score**           | 0.08                | **0.9558** | 0.92                        | 0.1895     | –                    | –          |
> | **Granger Causality** | –                   | –      | –                           | –          | 0.78                 | 0.6460     |
>
> We are extremely thankful for your detailed feedback and have further revised all editorial issues in the revised manuscript. We have also updated the limitation section to also touch on broader impact as well as current model & benchmark limitations. We have additionally revised the pie charts to stacked bar plots and changed the sample dataset to be in json format as suggested.

---

> > ### Comment · Reviewer_qH5M · 2026-02-03
> >
> > Thanks for the detailed response. I believe that my concerns are mostly addressed.
> >
> > Regarding causality, I thank the authors for more explicitly stating that Granger causality is considered in the manuscript. Still, claiming causality can be quite contentious, and I still believe it would be good to state this more generally when causality is mentioned -- the first time "Granger" is mentioned is on the bottom of page 6, after several brief discussions about the causal capabilities of TS-Reasoner.
> >
> > > All baseline agents have access to standard machine learning and data processing libraries (e.g., statsmodels, scikit-learn, Granger causality utilities).
> >
> > Thanks for the clarification. Though, unless I'm missing it (please let me know if I am), this information still doesn't appear in the manuscript. Please do include it, as I think this is quite important context.
> >
> > > [Updated Baselines]
> >
> > Thank you for the updated baselines, which I think improve the manuscript significantly. One remaining baseline (aside from the newly included foundation models) that I would be interested in seeing are AutoML-style workflows, like [AutoGluon](https://auto.gluon.ai/stable/index.html). This admittedly does not cover all the tasks that TS-Reasoner does, but is in some ways a more fair comparison.
> >
> > ### Minor Comments
> >
> > - One new typo was introduced on page 2 ("TS-Reasonerleverages" is missing a space).

---

### Decision · Action_Editor_Ri9B · 2026-02-25

**Recommendation:** Accept as is

**Audience:**

Yes

**Audience Explanation:**

Although reviewer qH5M remains concerned that the engineers applying the methods will have hard time digesting the more scientific spin of the narrative, there seems to be a consensus that the paper is useful for practicioners.

**Claims And Evidence:**

Yes

**Claims Explanation:**

All three reviewers have agreed with this statement, and reviewer 8oWC explicitly mentioned it in their overall final recommendation, while reviewer qH5M acknowledged the improvements in the paper after rebuttal.